# Ground-level gaseous pollutants ($NO_2$, $SO_2$, and CO) in China: daily seamless mapping and spatiotemporal variations

Jing Wei[1*], Zhanqing Li[1*], Jun Wang[2], Can Li[1], Pawan Gupta[3,4], Maureen Cribb[1]

1. Department of Atmospheric and Oceanic Science, Earth System Science Interdisciplinary Center, University of Maryland, College Park, MD, USA
2. Department of Chemical and Biochemical Engineering, Iowa Technology Institute, Center for Global and Regional Environmental Research, University of Iowa, USA
3. STI, Universities Space Research Association (USRA), Huntsville, AL, USA
4. NASA Marshall Space Flight Center, Huntsville, AL, USA

* Correspondence:

Zhanqing Li (zhanqing@umd.edu) and Jing Wei (weijing_rs@163.com; weijing@umd.edu)

**Abstract**

Gaseous pollutants at the ground level seriously threaten the urban air quality environment and public health. There are few estimates of gaseous pollutants that are spatially and temporally resolved and continuous across China. This study takes advantage of big data and artificial intelligence technologies to generate seamless daily maps of three major ambient pollutant gases, i.e., $NO_2$, $SO_2$, and CO, across China from 2013 to 2020 at a uniform spatial resolution of 10 km. Cross-validation between our estimates and ground observations illustrated a high data quality on a daily basis for surface $NO_2$, $SO_2$, and CO concentrations, with mean coefficients of determination (root-mean-square errors) of 0.84 (7.99 $\mu g/m^3$), 0.84 (10.7 $\mu g/m^3$), and 0.80 (0.29 $mg/m^3$), respectively. We found that the COVID-19 lockdown had sustained impacts on gaseous pollutants, where surface CO recovered to its normal level in China on around the 34[th] day after the Lunar New Year, while surface $SO_2$ and $NO_2$ rebounded more than twice slower due to more CO emissions from increased residents' indoor cooking and atmospheric oxidation capacity. Surface $NO_2$, $SO_2$, and CO reached their peak annual concentrations of $21.3 \pm 8.8$ $\mu g/m^3$, $23.1 \pm 13.3$ $\mu g/m^3$, and $1.01 \pm 0.29$ $mg/m^3$ in 2013, then continuously declined over time by 12%, 55%, and 17%, respectively, until 2020. The declining rates were more prominent from 2013 to 2017 due to the sharper reductions in anthropogenic emissions but have slowed down in recent years. Nevertheless, people still suffer from high-frequency risk exposure to surface $NO_2$ in eastern China, while surface $SO_2$ and CO have almost reached the recommended air quality guidelines level since 2018, benefiting from the implemented stricter "ultra-low" emission standards. This reconstructed dataset of surface gaseous pollutants will benefit future (especially short-term) air pollution and environmental health-related studies.

## 1. Introduction

Air pollution has been a major environmental concern, affecting human health, weather, and climate (Anenberg et al., 2022; Kan et al., 2012; Li et al., 2017a; Murray et al., 2020; Orellano et al., 2020), thus drawing worldwide attention. The sources of air pollution are complex. They include natural sources such as wildfires and anthropogenic emissions, including pollutants discharged from industrial production [e.g., smoke/dust, sulfur oxides, nitrogen oxides ($NO_x$), and volatile organic compounds (VOCs)], hazardous substances released from burning coal during heating seasons [e.g., dust, sulfur dioxide ($SO_2$), and carbon monoxide (CO)], and waste gases (e.g., CO, $SO_2$, and $NO_x$) generated by transportation, especially in big cities.

Among various air pollutants, the following have been most widely recognized: particulate matter with diameters smaller than 2.5 µm and 10 µm ($PM_{2.5}$ and $PM_{10}$) and gaseous pollutants [e.g., ozone ($O_3$), nitrogen dioxide ($NO_2$), $SO_2$, and CO, among others]. Many countries have built ground-based networks to monitor a variety of conventional pollutants in real time. China has experienced serious ambient air pollution for a long time, prompting the establishment of a large-scale air quality monitoring network (MEE, 2018a). Over the years, much effort has been made to model different species of air pollutants. Many studies focused on particulate matter in China have been carried out (Gao et al., 2022; Li et al., 2017b; Li et al., 2022b; Ma et al., 2022; Yang et al., 2022; Zhang et al., 2018). The global COVID-19 pandemic has motivated many attempts to estimate surface $NO_2$ concentrations from satellite-retrieved tropospheric $NO_2$ products (Tian et al., 2020; WHO, 2020), e.g., from the Ozone Monitoring Instrument (OMI) onboard the NASA Aura spacecraft and the TROPOspheric Monitoring Instrument (TROPOMI) onboard the Copernicus Sentinel-5 Precursor satellite, adopting different statistical regression (Chi et al., 2021; Qin et al., 2017; Zhang et al., 2018) and artificial intelligence (Chen et al., 2019; Chi et al., 2022; Dou et al., 2021; Liu, 2021; Wang et al., 2021; Zhan et al., 2018) models. By comparison, surface $SO_2$ and CO in China are less studied, limited by weaker signals and a lack of good-quality satellite tropospheric products (Han et al., 2022b; Li et al., 2020; Liu et al., 2019; Wang et al., 2021). Such studies still face more challenges, e.g., satellite data gaps and missing values that seriously limit their application and the neglect of spatiotemporal differences in air pollution in the modeling process. In

addition, most previous studies mainly focused on studying a single or a few species during
relatively short observational periods.

In view of the above problems, the purpose of this paper is to reconstruct daily concentrations of
three ambient gaseous pollutants (i.e., $NO_2$, $SO_2$, and CO) in China. To this end, relying on the
dense national ground-based observation network and big data, including satellite remote sensing
products, meteorological reanalysis, chemical model simulations, and emission inventories, we are
capable of mapping three pollutant gases seamlessly (100% spatial coverage) on a daily basis at a
uniform spatial resolution of 10 km since 2013 in China. Estimates were made using an extended
and powerful machine-learning model incorporating spatiotemporal information, i.e., space-time
extra-trees. Natural and anthropogenic effects on air pollution, including their physical mechanisms
and chemical reactions, were accounted for in the modeling. Using this dataset, spatiotemporal
variations of the gaseous pollutants, the impacts of environmental protection policies and the
COVID-19 epidemic, and population risk exposure to gaseous pollution are investigated.

To date, we have combined the advantages of artificial intelligence and big data to construct a
virtually complete set of major air quality parameters concerning both particulate and gaseous
pollutants over a long period of time across China, including $PM_1$ (1 km, 2000–Present) (Wei et al.,
2019), $PM_{2.5}$ (1 km, 2000–Present) (Wei et al., 2020; Wei et al., 2021a), $PM_{10}$ (1 km, 2000–Present)
(Wei et al., 2021b), $O_3$ (10 km, 1979–Present) (Wei et al., 2022a; He et al., 2022b), and $NO_2$ (1 km,
2019–Present) (Wei et al., 2022b), serving environmental, public health, economy, and other related
research. This study is the continuation of our previous studies, which adds two new species of $SO_2$
and CO for the first time and also dates the data records of $NO_2$ back to 2013. Instead of devoting
itself to a single pollutant, this study deals with all gaseous pollutants of compatible quality over the
same period with the same spatial coverage and resolution. In particular, considering that there are
few public datasets of these three gaseous pollutants with such spatiotemporal coverages focusing
on the whole of China, this is highly valuable for the sake of studying their variations, relative
proportions, and attribution of emission sources, as well as their diverse and joint effects of different
pollutant species on public health.

## 2. Materials and methods

## 2.1 Big data

### 2.1.1 Ground-based measurements

Hourly measurements of ground-level $NO_2$, $SO_2$, and CO concentrations from ~1600 reference-grade ground-based monitoring stations (Figure 1) collected from the China National Environmental Monitoring Centre (CNEMC) network were employed in the study. This network includes urban assessing stations, regional assessing stations, background stations, source impact stations, and traffic stations, set up in a reasonable overall layout that covers industrial (~14%), urban (~31%), suburban (~39%), and rural (~16%) areas to improve the spatial representations, continuity, and comparability of observations (HJ 664-2013) (MEE, 2013a). $NO_2$ is measured by chemiluminescence and differential optical absorption spectroscopy (DOAS), and $SO_2$ uses ultraviolet fluorescence and DOAS, while CO adopts non-dispersive infrared spectroscopy and gas filter correlation infrared spectroscopy. These measurements have been fully validated and have the same average error of indication of ±2% F.S. for the three gaseous pollutants considered here, with additional quality-control checks such as zero and span noise and zero and span drift (HJ 193-2013 and HJ 654-2013) (MEE, 2013b, 2013c). They have also been used as ground truth in almost all air pollutant modelling studies in China (Ma et al., 2022; Zhang et al., 2022a). All stations use the same technique to measure each gas routinely and continuously 24 hours a day at about the sea level without time series gaps. However, the reference state (i.e., observational conditions like temperature and pressure) changed from the standard condition (i.e., 273 K and 1013 hPa) to the room condition (i.e., 298 K and 1013 hPa) on 31 August 2018 (MEE, 2018a). We thus first converted observations of the three gaseous pollutants after this date to the uniform standard condition for consistency. Here, daily values for each air pollutant were averaged from at least 30% of valid hourly measurements at each station in each year from 2013 to 2020.

*[Please insert Figure 1 here]*

### 2.1.2 Main predictors

A new daily tropospheric $NO_2$ dataset at a horizontal resolution of $0.25° \times 0.25°$ in China was employed, created using a developed framework integrating OMI/Aura Quality Assurance for Essential Climate Variables (QA4ECV) and Global Ozone Monitoring Experiment–2B (GOME-2B) offline tropospheric $NO_2$ retrievals passing quality controls (i.e., cloud fraction < 0.3, surface albedo < 0.3, and solar zenith angle < 85°) (He et al., 2020). The reconstructed tropospheric $NO_2$ agreed well (R = 0.75–0.85) with Multi-AXis Differential Optical Absorption Spectroscopy (MAX-DOAS) measurements. Through this data fusion, the daily spatial coverage of satellite tropospheric $NO_2$ was significantly improved in China (average = 87%). Areas with a small number of missing values were imputed via a nonparametric machine-learning model by regressing the conversion relationship with Copernicus Atmosphere Monitoring Service (CAMS) tropospheric $NO_2$ assimilations ($0.75° \times 0.75°$), making sure that the interpolation was consistent with the OMI/Aura overpass time (Inness et al., 2019; Wang et al., 2020b). The gap-filled tropospheric $NO_2$ was reliable compared with measurements (R = 0.94–0.98) (Wei et al., 2022b). The above two-step gap-filling procedures allowed us to generate a daily seamless tropospheric $NO_2$ dataset that removes the effects of clouds from satellite observations.

Here, the reconstructed daily seamless tropospheric $NO_2$, together with CAMS daily ground-level $NO_2$ assimilations ($0.75° \times 0.75°$) averaged from all 3-hourly data in a day and monthly $NO_x$ anthropogenic emissions ($0.1° \times 0.1°$) (Inness et al., 2019), were used as the main predictors for estimating surface $NO_2$. Limited by the quality of direct satellite observations, daily model-simulated $SO_2$ and CO surface mass concentrations, averaged from all available data in a day provided by one-hourly Modern-Era Retrospective Analysis for Research and Applications, version 2 (MERRA-2, $0.625° \times 0.5°$), 3-hourly CAMS ($0.75° \times 0.75°$), and 3-hourly Goddard Earth Observing System Forward-Processing ($0.3125° \times 0.25°$) global reanalyses were used as main predictors to retrieve surface $SO_2$ and CO, together with CAMS monthly $SO_2$ and CO anthropogenic emissions.

### 2.1.3 Auxiliary factors

Meteorological factors have important diverse effects on air pollutants (He et al., 2017; Li et al., 2019), e.g., the boundary-layer height reflects their vertical distribution and variations (Li et al., 2017a; Seo et al., 2017); temperature, humidity, and pressure can affect their photochemical reactions (Li et al., 2019; Xu et al., 2011; Zhang et al., 2019a); and rainfall and wind can also influence their removal, accumulation, and transport (Dickerson et al., 2007; Li et al., 2019). Eight daily meteorological variables, provided by the ERA5-Land (0.1° × 0.1°) (Muñoz-Sabater et al., 2021) and ERA5 global reanalysis (0.25° × 0.25°) (Hersbach et al., 2020), were calculated (i.e., accumulated for precipitation and evaporation while averaged for the others) from all hourly data in a day, used as auxiliary variables to improve the modelling of gaseous pollutants. Other auxiliary remote-sensing data used to describe land-use cover/change [i.e., Moderate Resolution Imaging Spectroradiometer (MODIS) normalized difference vegetation index (NDVI), 0.05° × 0.05°] and population distribution density (i.e., LandScan$^{TM}$, 1 km) were employed as inputs to the machine-learning model because they are highly related to the type of pollutant emission and amounts of anthropogenic emissions, as well as the surface terrain [i.e., Shuttle Radar Topography Mission (SRTM) digital elevation model (DEM), 90m], which can affect the transmission of air pollutants. Table S1 provides detailed information about all the data used in this study. All variables were aggregated or resampled into a 0.1° × 0.1° resolution for consistency.

### 2.2 Pollutant gas modelling

Here, the developed Space-Time Extra-Tree (STET) model, integrating spatiotemporal autocorrelations of and differences in air pollutants to the Extremely Randomized Trees (ERT) (Wei et al., 2022a), was extended to estimate surface gaseous pollutants, i.e., $NO_2$, $SO_2$, and CO. ERT is an ensemble machine-learning model based on the decision tree, capable of solving the nonparametric multivariable nonlinear regression problem. Ensemble learning can avoid the lack of learning ability of a single learner, greatly improving accuracy. The introduced randomness enhances the model's anti-noise ability and minimizes the sensitivity to outliers and multicollinearity issues. It can handle high latitude, discrete or continuous data without data normalization and is easy to implement and parallel. However, several limitations exist, e.g., it is

difficult to make predictions beyond the range of training data, and there will be an over-fitting
issue on some regression problems with high noise. The training efficiency diminishes with
increasing memory occupation when the number of decision trees is large (Geurts et al., 2006).

Compared with traditional tree-based models (e.g., random forest), ERT has a stronger randomness
which randomly selects a feature subset at each node split and randomly obtains the optimal branch
attributes and thresholds. This helps to create more independent decision trees, further reducing
model variance and improving training accuracy (Geurts et al., 2006). The STET model has been
successfully applied in estimating high-quality surface $O_3$ in our previous study (Wei et al., 2022a).
It is thus extended here to regress the nonlinear conversion relationships between ground-based
measurements and the main predictors and auxiliary factors for other species of gaseous pollutants.
For surface $NO_2$, the STET model was applied to the main variables of the satellite tropospheric
$NO_2$ column, modelled surface $NO_2$ mass, and $NO_x$ emissions, together with ancillary variables of
the previously mentioned meteorological, surface, and population variables (Equation 1). For
surface $SO_2$ (Equation 2) and CO (Equation 3), modelled surface $SO_2$ and CO concentrations and
$SO_2$ and CO emissions were used as main predictors along with the same auxiliary variables as $NO_2$
to construct the STET models separately.

$NO_{2(ijt)} \sim f_{STET}(SNO_{2(ijt)}, MNO_{2(ijt)}, ENOx_{ijm}, Meteorology_{ijt}, NDVI_{ijm}, DEM_{ijy}, POP_{ijy}, P_s, P_t),$     (1)
$SO_{2(ijt)} \sim f_{STET}(MSO_{2(ijt)}, ESO_{2(ijm)}, Meteorology_{ijt}, NDVI_{ijm}, DEM_{ijy}, POP_{ijy}, P_s, P_t),$ (2)
$CO_{ijt} \sim f_{STET}(MCO_{ijt}, ECO_{ijm}, Meteorology_{ijt}, NDVI_{ijm}, DEM_{ijy}, POP_{ijy}, P_s, P_t),$   (3)

where $NO_{2(ijt)}$, $SO_{2(ijt)}$, and $CO_{ijt}$ indicate daily ground-based $NO_2$, $SO_2$, and CO measurements at
one grid $(i, j)$ on the $t$th day of a year; $SNO_{2(ijt)}$ indicates the daily satellite tropospheric $NO_2$ column
at one grid $(i, j)$ on the $t$th day of a year; $MNO_{2(ijt)}$, $MSO_{2(ijt)}$, and $MCO_{ijt}$ indicate daily model-
simulated surface $NO_2$, $SO_2$, and CO concentrations at one grid $(i, j)$ on the $t$th day of a year;
$ENOx_{ijm}$, $ESO_{2(ijm)}$, and $ECO_{ijm}$ indicate monthly anthropogenic $NO_x$, $SO_2$, and CO emissions at one
grid $(i, j)$ in the $m$th month of a year; $Meteorology_{ijt}$ represents each meteorological variable at one
grid $(i, j)$ on the $t$th day of a year; $DEM_{ijy}$ and $POP_{ijy}$ indicate the elevation and population at one
grid $(i, j)$ of a year; and $P_s$ and $P_t$ indicate the space and time terms (Wei et al., 2022a).

**3. Results and discussion**
**3.1 Seamless mapping of surface gaseous pollutants**
Using the constructed STET model, we generated daily 10 km resolution datasets with complete
coverage (spatial coverage = 100%) for three ground-level gaseous pollutants from 2013 to 2020 in
China, called ChinaHighNO$_2$, ChinaHighSO$_2$, and ChinaHighCO. Monthly and annual maps were
generated by directly averaging daily data at each grid. They belong to a series of public long-term,
full-coverage, high-resolution, and high-quality datasets of a variety of ground-level air pollutants
for China [ChinaHighAirPollutants (CHAP)] developed by our team. Figure 2 shows spatial
distributions of the three pollutant gases across China on a typical day (1 January 2018). The spatial
patterns of these gaseous pollutants were consistent with those observed on the ground, especially
in highly polluted areas, e.g., severe surface NO$_2$ pollution in the North China Plain (NCP) and high
surface SO$_2$ emissions in Shanxi Province. The unique advantage of our dataset is that it can
provide valuable gaseous pollutant information on a daily basis at locations in China where ground
measurements are not available. This addresses the major issues of scanning gaps and numerous
missing values in satellite remote sensing retrievals at cloudy locations, e.g., the average spatial
coverage of the official OMI/Aura daily tropospheric NO$_2$ product is only 42% over the whole of
China during the period 2013–2020 (Figure S1). Our dataset provides spatially complete coverage,
significantly increasing daily satellite observations by 58%. In addition, reanalysis data do not
simulate surface masses of gaseous pollutants well, underestimating them compared to our results
and ground-based observations in China (Figure S2). This is especially so for SO$_2$, where high-
pollution hot spots are easily misidentified. Validation illustrates that our regressed results for
surface NO$_2$, SO$_2$, and CO agree better with ground measurements than modelled results (slopes are
close to 1, and correlations > 0.93), 1.9–6.4 times stronger in slope, 1.3–3.5 times higher in
correlation, but 5.9–7.7 times smaller in differences (Figure S3). This shows that our model can take
advantage of big data to significantly correct and reconstruct gaseous simulation results via data
mining using machine learning.

236                    *[Please insert Figure 2 here]*

Figure 3 shows annual and seasonal maps for each gas pollutant during the period 2013–2020 across China. Multi-year mean surface $NO_2$, $SO_2$, and CO concentrations were $20.3 \pm 4.7$ μg/m$^3$, $16.2 \pm 7.7$ μg/m$^3$, and $0.86 \pm 0.22$ mg/m$^3$, respectively. Pollutant gases varied significantly in space across China, where high surface $NO_2$ levels were mainly distributed in typical urban agglomerations, e.g., the Beijing-Tianjin-Hebei (BTH) region, the Yangtze River and Pearl River Deltas (YRD and PRD), and scattered large cities with intensive human activities and highly developed transportation systems (e.g., Urumqi, Chengdu, Xi'an, and Wuhan, among others). High surface $SO_2$ concentrations were mainly observed in northern China (e.g., Shanxi, Hebei, and Shandong Provinces), associated with combustion emissions from anthropogenic sources, and the Yunnan Guizhou Plateau in southwest China, likely associated with emissions from volcanic eruptions. By contrast, except in some areas in central China (e.g., Shanxi and Hebei), surface CO concentrations were overall low.

Significant differences in spatial patterns were seen at the seasonal level. Surface $NO_2$, $SO_2$, and CO in summer (average = $15.9 \pm 4.7$ μg/m$^3$, $22.9 \pm 13.4$ μg/m$^3$, and $1.1 \pm 0.3$ mg/m$^3$, respectively) were the lowest, thanks to favorable meteorological conditions, e.g., abundant precipitation and high air humidity conducive to flushing and scavenging of different air pollutants (Yoo et al., 2014). Strong sunlight and high temperature also accelerate the photochemical reactions of $NO_2$ loss (Shah et al., 2020). Pollution levels were highest in winter, with average values increasing by ~1.5–1.9 times those in summer. This difference was much larger in central and eastern China, e.g., 2.3–3.4 times higher in the BTH due to large amounts of direct $NO_x$, $SO_2$, and CO emissions from burning coal for heating in winter in northern China. The spatial patterns of the three gaseous pollutants were similar in spring and autumn.

*[Please insert Figure 3 here]*

**3.2 Changes in gaseous pollution and exposure risk**

**3.2.1 Short-term epidemic effects on air quality**

Many studies have focused on the effects of the COVID-19 epidemic on air quality (WHO, 2020). Most of them were done using ground-based observations (Huang et al., 2020; Su et al., 2020),

tropospheric gas columns (Field et al., 2021; Levelt et al., 2022), or retrieved surface masses
(Cooper et al., 2022; Ling and Li, 2021). The resulting conclusions could be affected by insufficient
spatial representation due to the uneven distribution of ground monitors or a large number of
missing values in space due to the influence of clouds. The unique advantage of our seamless day-
to-day gaseous pollutant dataset can make up for these shortcomings, allowing us to assess the
changes more accurately and quantitatively in gaseous pollutants during the epidemic.

We first compared the spatial differences in monthly relative differences from February to April
between 2020 and 2019 in China (Figure 4). In February, surface $NO_2$ sharply reduced in China,
especially in key urban agglomerations and megacities, showing relative changes of greater than
50%. A significant decrease in surface $SO_2$ (> 40%) was observed in northern areas where heavy
industry is the mainstay in China (e.g., Tianjin, Hebei, and Shandong), while little change was seen
in southern China. Surface CO also showed drastic decreases, but the amplitude was smaller than
the other two gaseous pollutants. These were attributed to extensive plant closures and traffic
controls due to the lockdown, which started at the end of January 2020, significantly reducing
anthropogenic $NO_x$, $SO_2$, and CO emissions (Ding et al., 2020; Yang et al., 2022; Zheng et al.,
2021). In March, surface $NO_2$ was still generally lower than the historical level in most eastern
areas, especially in areas where the epidemic was severe, i.e., Wuhan, Hubei Province, and its
surrounding areas. The decrease in surface $SO_2$ largely slowed by more than two times in the NCP
and central China, while surface CO almost returned to normal levels in most areas in China. In
April, surface $NO_2$ and $SO_2$ were comparable to historical concentrations (within $\pm$ 10%), even
increasing in some areas of southern and northeastern areas due to rebounding anthropogenic
emissions (Ding et al., 2020), especially in Hubei Province, indicating that their surface levels were
almost recovered.

289                          *[Please insert Figure 4 here]*

Most previous studies have focused mainly on changes during the lockdown, with little attention
paid to the recovery. We thus compared the time series of daily population-weighted concentrations
of the three gaseous pollutants after the Lunar New Year between 2020 and 2019 in China (Figure

5). After the beginning of New Year's Eve, surface gaseous pollutants showed a significant decrease in both the normal and epidemic years due to the closure of factories, with decreasing anthropogenic emissions during the Spring Festival holiday. However, gaseous pollutants in the normal year rose rapidly after they fell to their lowest levels due to the return to work after the holidays. By contrast, their levels continued to decrease in 2020 and were lower than historical levels due to the sustained impacts of the strict lockdowns. They hit bottom in the 4th week after the Lunar New Year, then began to increase gradually. Surface $NO_2$ and $SO_2$ recovered in the middle of the 11th week (around the 72nd and 75th days) after the Lunar New Year (i.e., 2020 and 2019 concentrations intersected and then alternately changed). However, surface CO levels recovered at the end of the 5th week (around the 34th day), more than twice faster than $NO_2$ and $SO_2$ levels. This is attributed to more CO emissions from increased residents' indoor cooking (Zheng et al., 2018), increased atmospheric oxidation capacity (Huang et al., 2020; Wei et al., 2022a), and a potentially higher sensitivity to temperature rises (Lin et al., 2021).

*[Please insert Figure 5 here]*

### 3.2.2 Temporal variations and policy implications

Figures S4-S6 show annual mean maps of each gaseous pollutant from 2013 to 2020 in China. Surface $NO_2$, $SO_2$, and CO changed greatly, peaking in 2013, with average values of $21.3 \pm 8.8$ $\mu g/m^3$, $23.1 \pm 13.3$ $\mu g/m^3$, and $1.01 \pm 0.29$ $mg/m^3$, respectively. They reached their lowest levels in 2020, particularly due to the noticeable effects of the COVID-19 epidemic. In general, national ambient $NO_2$, $SO_2$, and CO concentrations decreased by approximately 12%, 55%, and 17% from 2013 to 2020, respectively. Large seasonal differences were observed in the amplitude of gaseous pollutant (Figure 6), e.g., surface $NO_2$ decreased the most in winter, especially in the three urban agglomerations ($\downarrow$24–31%), changing the least in autumn (especially in the YRD). Surface $SO_2$ showed much larger decreases in all seasons, especially during the cold seasons ($\downarrow$55–81%), due to the implementation of stricter "ultra-low" emission standards (Li et al., 2022a; Zhang et al., 2019b). Surface CO had similar seasonal changes as $SO_2$ but 1.5–3.3 times smaller in amplitude.

*[Please insert Figure 6 here]*

To better investigate the spatiotemporal variations of ambient gaseous pollution, we calculated
linear trends and significance levels using monthly anomalies by removing seasonal cycles. Most of
China showed significant decreasing trends, with average annual rates of 0.23 µg/m$^3$, 2.01 µg/m$^3$,
and 0.05 mg/m$^3$ for surface $NO_2$, $SO_2$, and CO ($p < 0.001$), respectively (Figure 7), especially in
three urban agglomerations and large cities (e.g., Wuhan and Chengdu). The largest downward
trends mainly occurred in northern and central China, especially in the BTH (Table 3). This is
mainly due to the change in fuel for heating from coal to gas widespread across China in winter
(Wang et al., 2020a), greatly reducing emissions of precursor gases (Koukouli et al., 2018).
Increasing trends of surface $NO_2$ were, however, found in Ningxia and Shanxi Provinces in central
China due to increased traffic emissions and new coal-burning power plants in underdeveloped
areas without strict regulations on $NO_x$ emissions (Li et al., 2022a; Maji and Sarkar, 2020; Van Der
A et al., 2017).

We then divided the study period into three periods to investigate the impact of major
environmental protection policies on air quality implemented in China (Figure 7). During the Clear
Air Action Plan (CAAP, 2013–2017), the rates of decrease for surface $NO_2$, $SO_2$, and CO
accelerated in most populated areas in China, especially urban areas. This was due to dramatic
reductions in main pollutant emissions like $SO_2$ and $NO_x$ (by 59% and 21%, respectively) through
the upgrading of key industries, industrial structure adjustments, and coal-fired boiler remediation
(Zhang et al., 2019b). In addition, the majority of gaseous pollutants had dropped continuously
during the Blue Sky Defense War (BSDW, 2018–2020), benefiting from continuous reductions in
total air pollutant emissions and the impacts of COVID-19 (Jiang et al., 2021; Zheng et al., 2021).
However, areas with trends passing the significance level sharply shrank, especially for surface $SO_2$.

During the 13$^{th}$ Five-Year-Plan (FYP, 2016–2020), the decreasing trends of the three gaseous
pollutants across China slowed down compared to those during CAAP. Large decreases in surface
$NO_2$ were mainly found in the BTH region and Henan Province, while slightly increasing trends
occurred in southern China. Surface $SO_2$ significantly decreased in most areas, where a greater
downward trend was observed in Shanxi Province, mainly due to the reduction in coal consumption

thanks to a strengthened clean-heating policy (Lee et al., 2021). Surface CO also continuously decreased, more rapidly in central China but less rapidly elsewhere. The continuous decline in gaseous pollutants is due to the binding reductions in total emissions of major pollutants like $NO_x$ ($\downarrow71\%$) and $SO_2$ ($\downarrow48\%$) in China (Wan et al., 2022; Wu et al., 2022c).

*[Please insert Figure 7 here]*

### 3.2.3 Population-risk exposure to gaseous pollution

With the daily seamless datasets, we can evaluate the spatial and temporal variations of short-term population-risk exposure to the three gaseous pollutants by calculating the number of days in a given year exceeding the new recommended short-term minimum interim target (IT1) and desired air quality guidelines (AQG) level defined by the WHO in 2021 (WHO, 2021). The area exceeding the recommended levels (i.e., daily $NO_2 > 120$ $\mu g/m^3$, $SO_2 > 125$ $\mu g/m^3$, and $CO > 7$ $mg/m^3$) was generally small in eastern China (Figure S7). High $NO_2$-exposure risks were mainly found in Beijing and Hebei Province and a handful of big cities (e.g., Jinan, Wuhan, Shanghai, and Guangzhou), while high $SO_2$-exposure risks were mainly observed in Hebei, Shandong, and Shaanxi Provinces. The risk of high CO pollution was small, only found in some scattered areas in the NCP. In general, both the area and the possibility of occurrence exposure to high pollution has gradually decreased over time, almost disappearing since 2018.

By contrast, most areas of eastern China had a surface $NO_2$ exposure exceeding the AQG level (Figure 8), especially in the north and economically developed areas in the south (proportion > 80%). Both the extent and intensity are decreasing over time, but it is still a problem, suggesting that stronger $NO_x$ controls are needed in the future. Most of the main air pollution transmission belt in China (i.e., the "2 + 26" cities, Figure 1) had surface $SO_2$ levels exceeding the AQG level at the beginning of the study period. Thanks to strict control measures, these polluted areas sharply decreased after 2015, almost disappearing in 2020. Controlling CO was much more successful in China, with less than 10% of the days in the BTH exceeding the acceptable standard in the early part of the study period. Most areas have reached the CO AQG level since 2018.

*[Please insert Figure 8 here]*

Figure 9 shows the percentage of days with pollution levels exceeding WHO air quality standards in
three key regions. BTH was the only region experiencing high $NO_2$ and $SO_2$ exposure risks (i.e.,
daily mean > IT1), dropping to zero since 2017 and 2016, while YRD and PRD had no high risks of
exposure to the three gaseous pollutants (Figure 9a-b). There was also no regional high CO-
pollution risk (Figure 9c). However, although declining continuously, regional surface $NO_2$ levels
failed to meet the short-term AQG level in 2020, with 61–73% of the days exceeding the AQG level.
More efforts toward mitigating $NO_2$ levels in these key regions are thus needed. Continual
decreases in the number of days above the AQG level were also observed in surface $SO_2$, reducing
to near zero in 2014, 2016, and 2018 in the PRD, YRD, and BTH, respectively. Less than 3% of the
days in the BTH and YRD had surface CO levels exceeding the AQG level. Surface CO levels were
always below the AQG level in the PRD.
*[Please insert Figure 9 here]*
**3.3 Data quality assessment**
Here, the widely used out-of-sample 10-fold cross-validation (10-CV) method was adopted to
evaluate the overall estimation accuracy of gaseous pollutants (Rodriguez et al., 2010; Wei et al.,
2022a). An additional out-of-station 10-CV approach was used to validate the prediction accuracy
of gaseous pollutants, performed based on measurements from ground monitoring stations. These
measurements were randomly divided into ten subsets, of which data samples from nine subsets
were used for model training and the remaining subset for model validation. This was done 10 times,
in turn, to ensure that data from all stations were tested. This procedure generates independent
training samples and test samples made in different locations, used to indicate the spatial prediction
ability of the model in areas where ground-based measurements are unavailable (Wei et al., 2022a;
Wu et al., 2021).

**3.3.1 Estimate and prediction accuracy**
Figure 10 shows the CV results of all daily estimates and predictions for ground-level $NO_2$, $SO_2$,
and CO concentrations from 2013 to 2020 in China (sample size: $N \approx 3.6$ million). Surface $NO_2$
and $SO_2$ concentrations mainly fell in the range of 200 to 500 $\mu g/m^3$. Daily estimates were highly

correlated to observations, with the same coefficients of determination ($R^2 = 0.84$) and slopes close

to 1 (0.86 and 0.84, respectively). Average root-mean-square error (RMSE) [mean absolute error

(MAE)] values of surface $NO_2$ and $SO_2$ estimates were 7.99 (5.34) and 10.07 (4.68) µg/m³, and

normalized RMSE (NRMSE) values were 0.25 and 0.51, respectively. Most daily CO observations

were less than 10 mg/m³, agreeing well with our daily estimates ($R^2 = 0.80$, slope = 0.79), and the

average RMSE (MAE) and NRMSE values were 0.29 (0.16) mg/m³ and 0.3. Compared to

estimation accuracies (Figure 10a-c), prediction accuracies slightly decreased, which is acceptable

considering the weak signals of trace gases. Daily surface $SO_2$, $NO_2$, and CO predictions (Figure

10d-f) agree well with ground measurements, with spatial $R^2$ values of 0.70, 0.68, and 0.61,

respectively. Their respective RMSE (MAE) values were 14.28 (8.1) µg/m³, 11.57 (7.06) µg/m³,

and 0.42 (0.24) mg/m³, and NRMSE values were 0.35, 0.71, and 0.42, respectively, representing the

accuracy for areas without ground monitoring stations.

*[Please insert Figure 10 here]*

The performance of our air pollution modelling was also evaluated on an annual basis, showing that

our model works well in estimating and predicting the concentrations of different surface gaseous

pollutants in different years (Table 1). The model performance has continuously improved over time,

as indicated by increasing correlations and decreasing uncertainties. This is because of the

increasing density of ground stations (especially in the suburban areas of cities) and updated quality

control of measurements, e.g., improving the sampling flow calibration of monitoring instruments,

flow calibration of dynamic calibrators, and revision of precision/accuracy review and data validity

judgment (HJ 818-2018) (MEE, 2018b). This has led to an increase in the number of data samples

(e.g., from 169 thousand in 2013 to more than 522 thousand in 2020) and improvement in their

quality.

*[Please insert Table 1 here]*

Figure 11 shows the spatial validation of estimated daily pollutant gases across China. In general,

our model works well at the site scale, with average CV-$R^2$ values of 0.77, 0.72, and 0.72, and

NRMSE values of 0.25, 0.43, and 0.26 for surface $NO_2$, $SO_2$, and CO, respectively. In addition,

approximately 93%, 80%, and 84% of the stations had at least moderate agreements (CV-$R^2 > 0.6$) between our estimates and ground measurements. Except for some scattered sites, the estimation uncertainties were generally less than 0.3, 0.5, and 0.3 in more than 80%, 77%, and 76% of the stations for the above three gaseous pollutant species, respectively.

*[Please insert Figure 11 here]*

Figure 12 shows the temporal validation of ground-level gaseous pollutants as a function of ground measurements in China. On the monthly scale (Figure 12a-c), we collected a total of ~119,000 matched samples of the three gaseous pollutants. Accuracies significantly improved, with increasing $R^2$ (decreasing RMSE) values of 0.93 (4.41 $\mu g/m^3$), 0.97 (4.03 $\mu g/m^3$), and 0.94 (0.13 $mg/m^3$) for surface $NO_2$, $SO_2$, and CO, respectively. On the annual scale (Figure 12d-f), more than ~10,000 matched samples were collected, showing better agreement with observations (e.g., $R^2 = 0.94$, 0.98, and 0.97) and lower uncertainties (e.g., RMSE = 3.06 $\mu g/m^3$, 2.46 $\mu g/m^3$, and 0.07 $mg/m^3$) for the above three gaseous pollutants, respectively.

*[Please insert Figure 12 here]*

**3.3.2 Comparison with previous studies**

We compared our results with those from previous studies on the estimation of the three gaseous pollutants using different developed models focusing on the whole of China. Here, only those studies applying the same out-of-sample cross-validation approach against ground-based measurements collected from the same CNEMC network were selected (Table 2). The statistics shown in the table come from the publications themselves because their generated datasets are not publicly available. We have applied the same validation method and ground measurements as those used in the previous studies. Most generated surface $NO_2$ datasets had numerous missing values in space limited by direct OMI/Aura satellite observations at spatial resolutions from 0.125°× 0.125° to 0.25°×0.25° (Chen et al., 2019; Chi et al., 2021; Dou et al., 2021; Xu et al., 2019; Zhan et al., 2018). Some studies improved the spatial resolution by introducing $NO_2$ data from the recently launched Sentinel-5 TROPOMI satellite, but data are only available from October 2018 onward (Chi et al., 2022; Liu, 2021; Wang et al., 2021; Wei et al., 2022b). Surface $SO_2$ estimated from an

SO₂ emission inventory and surface CO from Measurement of Pollution in the Troposphere (MOPITT) and TROPOMI retrievals have a much lower data quality, with smaller $R^2$ values by 12–57% and larger RMSE values by 41–47% against ground measurements compared to ours (Li et al., 2020; Liu et al., 2019; Wang et al., 2021). Overall, our gaseous pollutant datasets are superior to those from previous studies in terms of overall accuracy, spatial coverage, and length of data records.

*[Please insert Table 2 here]*

**3.4 Successful applications**

Our surface gaseous pollutant datasets have been freely available to the public online since March 2021 (See data availability). A large number of studies have used the three gaseous pollutant datasets generated in this study to study their single or joint impacts on environmental health from both long-term and short-term perspectives, benefiting from the unique daily spatially seamless coverage. For example, a nearly linear relationship between long-term ambient $NO_2$ and adult mortality in China was observed (Zhang et al., 2022b); ambient $NO_2$ hindered the survival of middle-aged and elderly people (Wang et al., 2023) while acute exposure to ambient $SO_2$ increased the risk of asthma mortality in China (Li et al., 2023b; Liu et al., 2022b; Liu et al., 2023). Long-term $SO_2$ and CO exposure can increase the incidence rate of visual impairment in children in China (Chen et al., 2022a), and short-term exposure to ambient CO can significantly increase the probability of hospitalization for stroke sequelae (Wang et al., 2022b). Regional and national cohort studies have shown that exposure, especially short-term exposure, to multiple ambient gaseous ($NO_2$, $SO_2$, and CO) and particulate pollutants have negative effects of varying degrees on a variety of diseases, like all-cause mortality (Feng et al., 2023), dementia mortality (Liu et al., 2022a), myocardial infarction mortality (Ma et al., 2023), cause-specific cardiovascular disease (Xu et al., 2022a; Xu et al., 2022b), respiratory diseases (Li et al., 2023a), ischemic and hemorrhagic stroke (Cai et al., 2022; He et al., 2022a; Wu et al., 2022b; Xu et al., 2022c), metabolic syndrome (Guo et al., 2022; Han et al., 2022a), influenza-like illness (Lu et al., 2023), incident dyslipidemia (Hu et al., 2023), diabetes (Mei et al., 2023), blood pressure (Song et al., 2022; Wu et al., 2022a), renal/ kidney function (Li et al., 2022c; Li et al., 2023c), neurodevelopmental delay (Su et al., 2022), serum liver

enzymes (Li et al., 2022d), overweight and obesity (Chen et al., 2022b), insomnia (Xu et al., 2021),
and sleep quality (Wang et al., 2022a). These studies attest well to the value of the CHAP dataset
regarding current and future public health issues, among others.

**4. Summary and conclusions**
Exposure to gaseous pollution is detrimental to human health, a major public concern in heavily
polluted regions like China, where ground-based observations are not as rich as in major developed
countries. Moreover, pollutants travel long distances, affecting large downstream regions. To
remedy such limitations, this study applied the machine-learning model called Space-Time Extra-
Tree to estimate ambient gaseous pollutants across China, with extensive input variables measured
by monitors and satellites, and models. Daily 10 km resolution (approximately $0.1° \times 0.1°$) seamless
(spatial coverage = 100%) datasets for ground-level $NO_2$, $SO_2$, and CO concentrations in China
from 2013 to 2020 were generated. These datasets were cross-evaluated in terms of overall
accuracy and predictive ability at different spatiotemporal levels. National daily estimates
(predictions) of surface $NO_2$, $SO_2$, and CO were highly consistent with ground measurements, with
average out-of-sample (out-of-station) CV-$R^2$ values of 0.84 (0.68), 0.84 (0.7), and 0.8 (0.61), and
RMSEs of 7.99 (11.57) $\mu g/m^3$, 10.7 (14.28) $\mu g/m^3$, and 0.29 (0.42) $mg/m^3$, respectively.

Ambient pollutant gases varied significantly in space and time, with high levels mainly found in the
North China Plain, especially in winter, due to more anthropogenic emissions, such as coal burning
for heating. All gaseous pollutants sharply declined in China during the COVID-19 outbreak, while
large differences were observed during their recovery times. For example, surface CO was the first
to return to its historical level within the fifth week after the Lunar New Year in 2020, about twice
faster as surface $NO_2$ and $SO_2$ levels. This is attributed to more home cooking and enhanced
atmospheric oxidation. Temporally, surface $NO_2$, $SO_2$, and CO levels in China gradually decreased
from peaks in 2013 (average = $21.3 \pm 8.8$ $\mu g/m^3$, $23.1 \pm 13.3$ $\mu g/m^3$, and $1.01 \pm 0.29$ $mg/m^3$,
respectively), with annual rates of decrease of 0.23 $\mu g/m^3$, 2.01 $\mu g/m^3$, and 0.05 $mg/m^3$,
respectively ($p < 0.001$), until 2020. Improvements in air quality have been made in the last eight
years, thanks to the implementation of a series of environmental protection policies, greatly

reducing pollutant emissions. In addition, both the areal extents of regions experiencing gaseous

pollution and the probability of gaseous pollution occurring have gradually decreased over time,

especially for surface CO and $SO_2$, which have almost reached the short-term air quality guidelines

level recommended by the WHO in most areas in China in 2020. This high-quality daily seamless

dataset of gaseous pollutants will benefit future environmental and health-related studies focused on

China, especially studies investigating short-term air pollution exposure.

Although a lot of new and/or useful data and analyses are presented in this study, they still suffer

from some limitations. For example, our estimated surface $SO_2$ and CO concentrations should have

larger uncertainties than those of $NO_2$ since model simulations stead of satellite retrievals are

supplemented during modelling to compensate for the lack of data in China. However, these data

often have large biases in the remote regions with few observations as in western China (Li et al.,

2022b), as the surface measurements from MEE are mainly over eastern China. More influential

factors stemming from regional economic and development differences, and more parameters

describing the complex meteorological system (e.g., winds at 850 hPa and the pressure system in

the mid-troposphere) need to be considered in developing more powerful artificial intelligence

models, which could be helpful in improving the accuracy of air pollutant retrievals. The

spatiotemporal resolutions of gaseous pollutants will be further improved by integrating information

from polar-orbiting and geostationary satellites to investigate diurnal variations. In a future study,

we will also reconstruct data records over the last two decades and investigate their long-term

spatiotemporal variations, filling the gap of missing observations. This will help us understand their

formation mechanisms and impacts on fine particulate matter and ozone pollution in China.

**Data availability**

CNEMC measurements of gaseous pollutants are available at http://www.cnemc.cn. The

reconstructed OMI/Aura tropospheric $NO_2$ product is available at

https://doi.org/10.6084/m9.figshare.13126847. MODIS series products and the MERRA-2

reanalysis are available at https://search.earthdata.nasa.gov/. The SRTM DEM is available at

https://www2.jpl.nasa.gov/srtm/, and LandScan[TM] population information is available at

https://landscan.ornl.gov/. The ERA5 reanalysis is available at https://cds.climate.copernicus.eu/,
GEOS CF data are available at https://portal.nccs.nasa.gov/datashare/gmao/, and the CAMS
reanalysis and emission inventory are available at https://ads.atmosphere.copernicus.eu/.

The ChinaHighAirPollutants (CHAP) dataset is open access and freely available at https://weijing-
rs.github.io/product.html. The ChinaHighNO$_2$ dataset is available at
https://doi.org/10.5281/zenodo.4641542, the ChinaHighSO$_2$ dataset is available at
https://doi.org/10.5281/zenodo.4641538, and the ChinaHighCO dataset is available at
https://doi.org/10.5281/zenodo.4641530.

**Author contributions**

JiW and ZL designed the study. JiW performed the research and wrote the initial draft of this paper.
ZL, JuW, CL, and PG reviewed and edited the paper. MC copyedited the article. All authors made
substantial contributions to this work.

**Competing interests**

The authors declare that they have no conflict of interest.

**Acknowledgments**

JiW, ZL, and JuW were supported by NASA Earth Sciences' Applied Science Programs
(80NSSC21K1980 and 80NSSC19K0950). PG was supported by NASA's Research Opportunities
in Space and Earth Science (ROSES-2020), Program Element A.38: Health and Air Quality
Applied Sciences Team.

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

**Figures**

**Figure 1.** Geographical locations of ground-based stations from the China National Environmental Monitoring Centre network (marked as yellow dots) monitoring gaseous pollutants across China. The background shows the nighttime-light level, an estimate of population. Purple boundaries three typical urban agglomerations: the Beijing-Tianjin-Hebei (BTH) region, the Yangtze River Delta (YRD), and the Pearl River Delta (PRD).

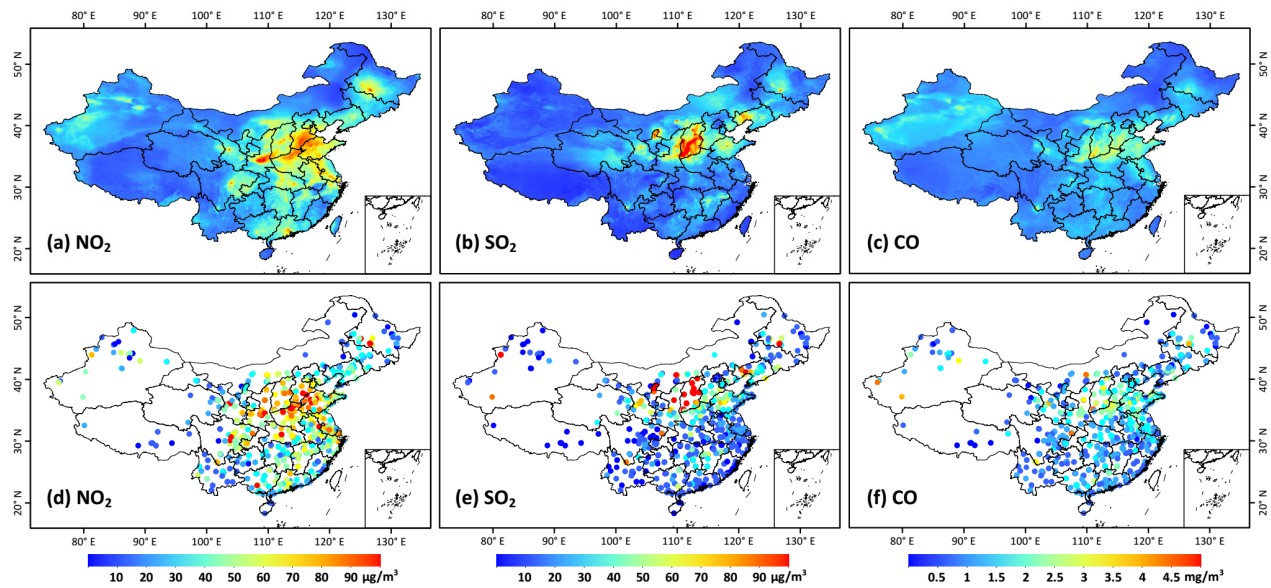

**Figure 2.** A typical example of (a-c) big-data-derived (horizontal resolution = 10 km) seamless surface NO₂ (μg/m³), SO₂ (μg/m³), and CO (mg/m³) concentrations and (d-f) corresponding ground measurements on 1 January 2018 in China.

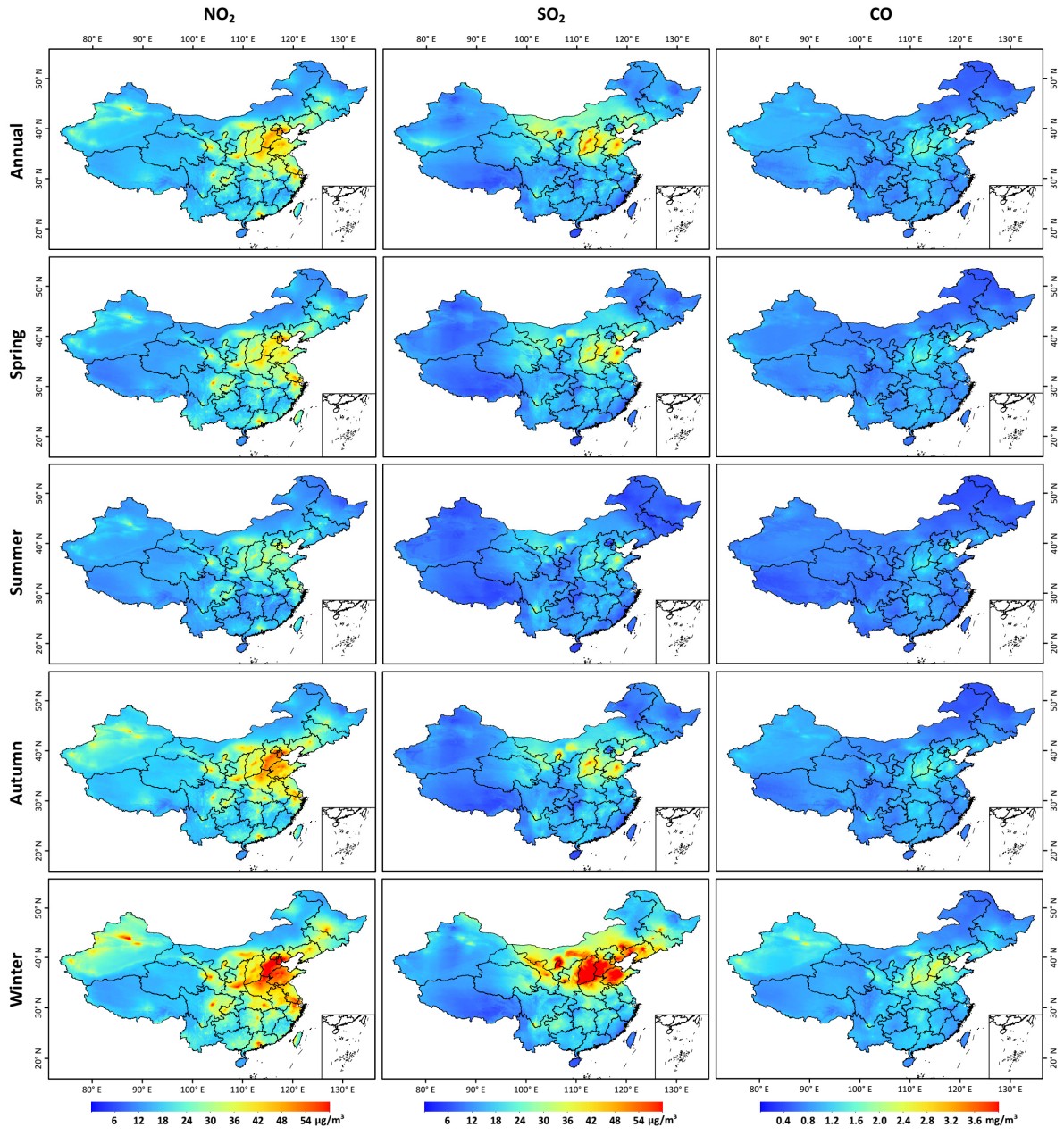

992

**Figure 3.** Annual and seasonal mean maps (horizontal resolution = 10 km) of surface $NO_2$ ($\mu g/m^3$),
993

SO$_2$ ($\mu g/m^3$), and CO ($mg/m^3$) averaged over the period 2013–2020 in China.
994

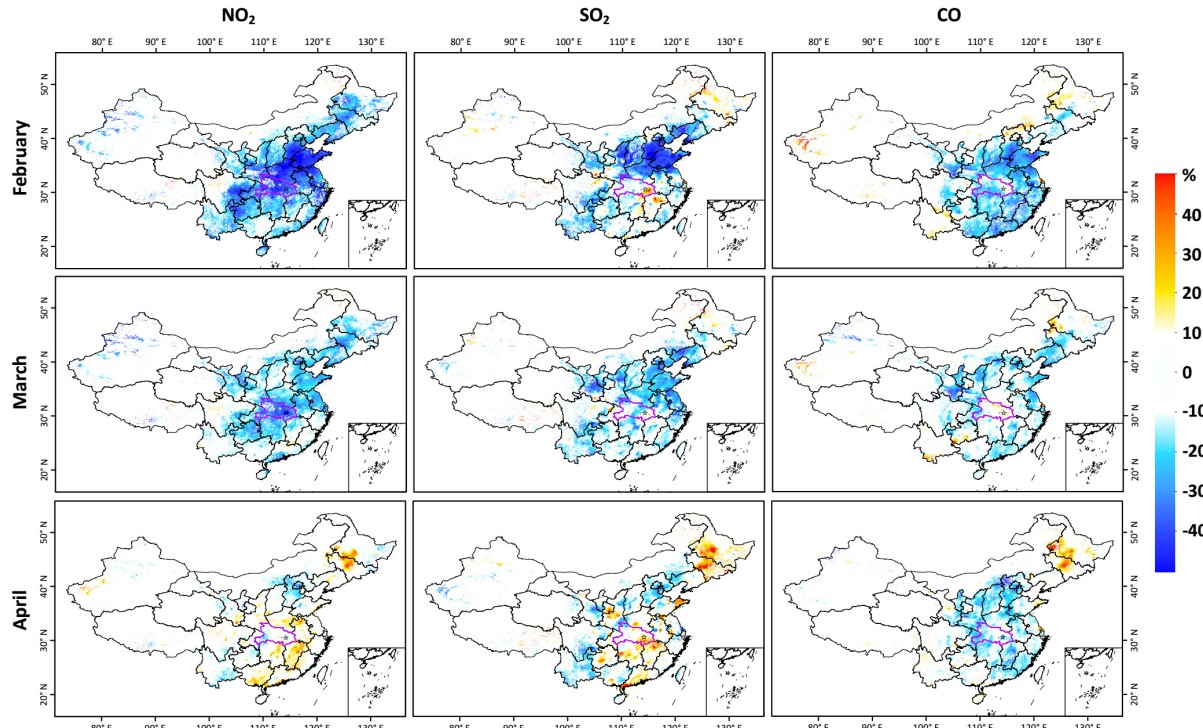

995
**Figure 4.** Relative changes (%) in surface $NO_2$, $SO_2$, and CO concentrations in February, March,
and April between 2019 and 2020 in populated areas of China. The area outlined in magenta and the
star in each panel indicate Hubei Province and Wuhan City, respectively.

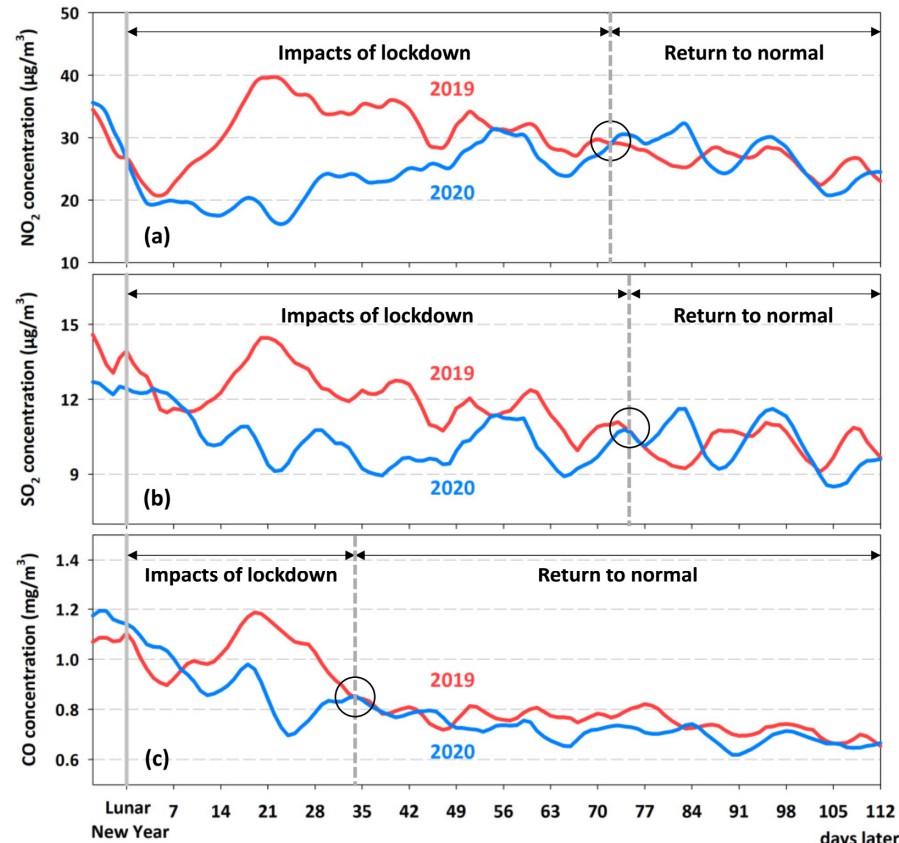

**Figure 5.** Time series of the seven-day moving averages of daily population-weighted surface (a) NO$_2$, (b) SO$_2$, and (c) CO concentrations after the Lunar New Year of 2019 and 2020 in China. The black circle in each panel shows the turning point when the gaseous pollutants began to return to their normal levels.

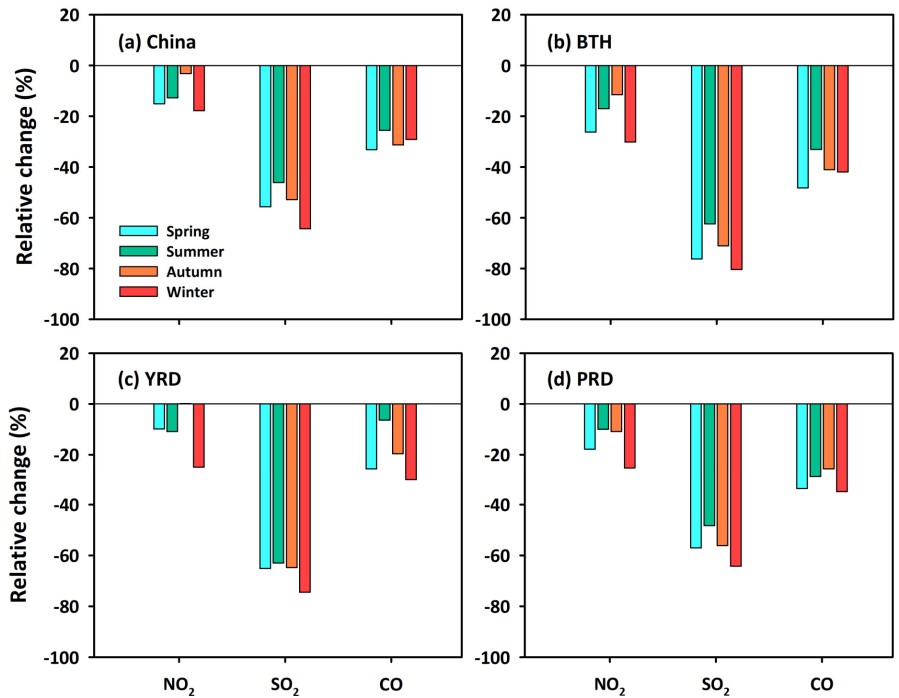

1006

**Figure 6.** Relative changes (%) in seasonal mean surface $NO_2$, $SO_2$, and CO concentrations
between 2013 and 2020 over (a) China, (b) the Beijing-Tianjin-Hebei (BTH) region, (c) the Yangtze
River Delta (YRD), and (d) the Pearl River Delta (PRD).

1007
1008
1009
1010

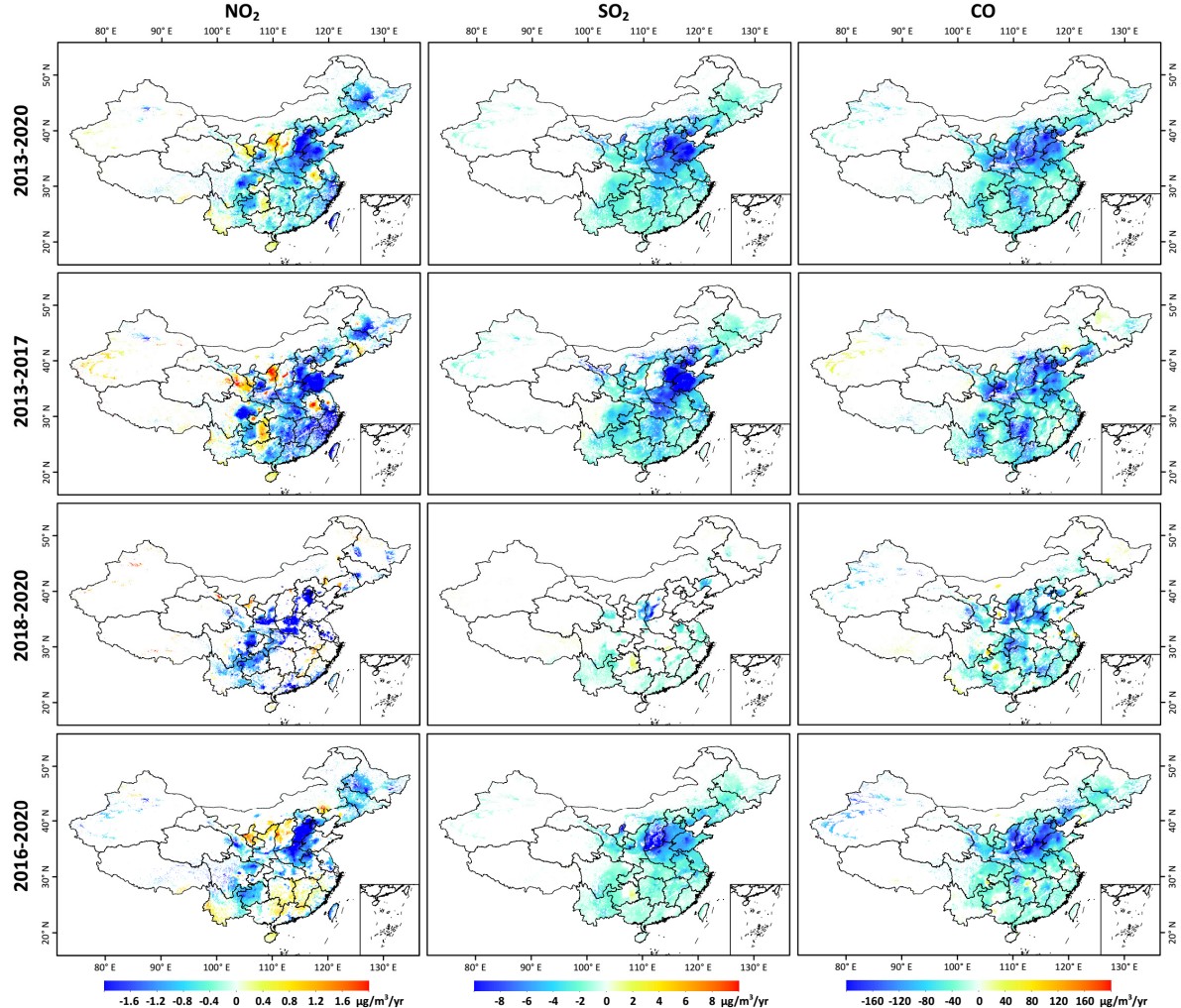

**Figure 7.** Temporal trends of surface $NO_2$, $SO_2$, and CO concentrations during the whole period (2013–2020), the Clean Air Action Plan (2013–2017), the Blue Sky Defense War (2018–2020), and the 13rd Five-Year Plan (2016–2020) in populated areas of China. Only regions with trends that are significant at the 95% ($p < 0.05$) confidence level are shown.

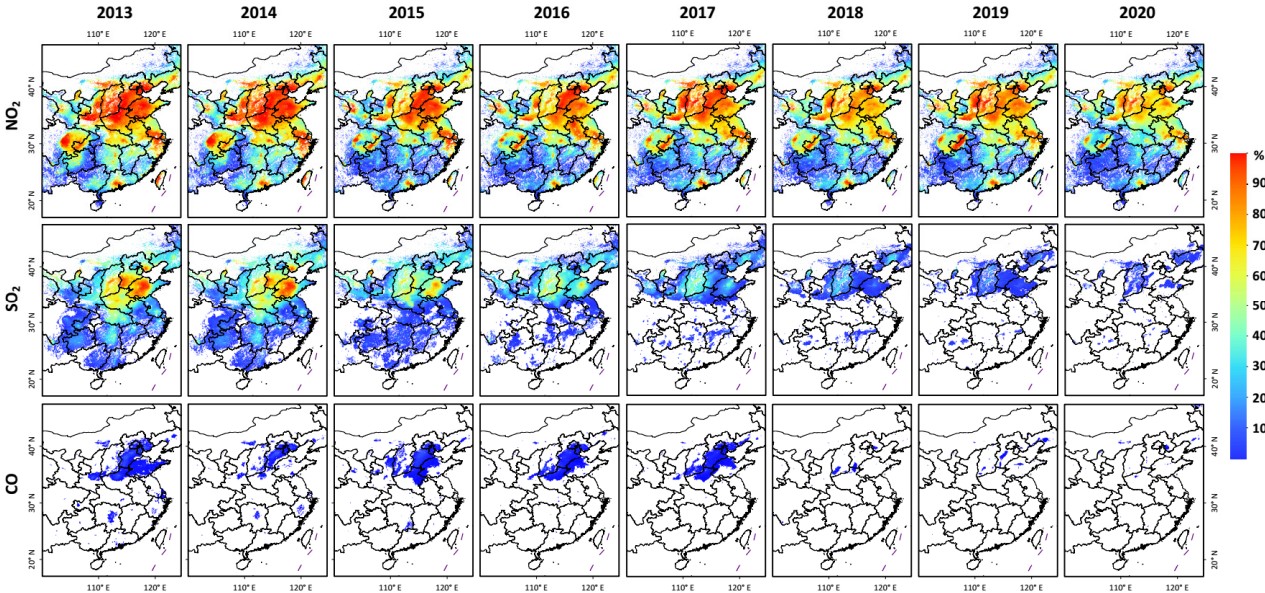

**Figure 8.** Spatial distributions of the percentage of days exceeding the WHO recommended short-term desired air quality guidelines (AQG) level for surface NO$_2$ (daily mean > 25 μg/m$^3$), SO$_2$ (daily mean > 40 μg/m$^3$), and CO (daily mean > 4 mg/m$^3$) for each year from 2013 to 2020 in populated areas of eastern China.

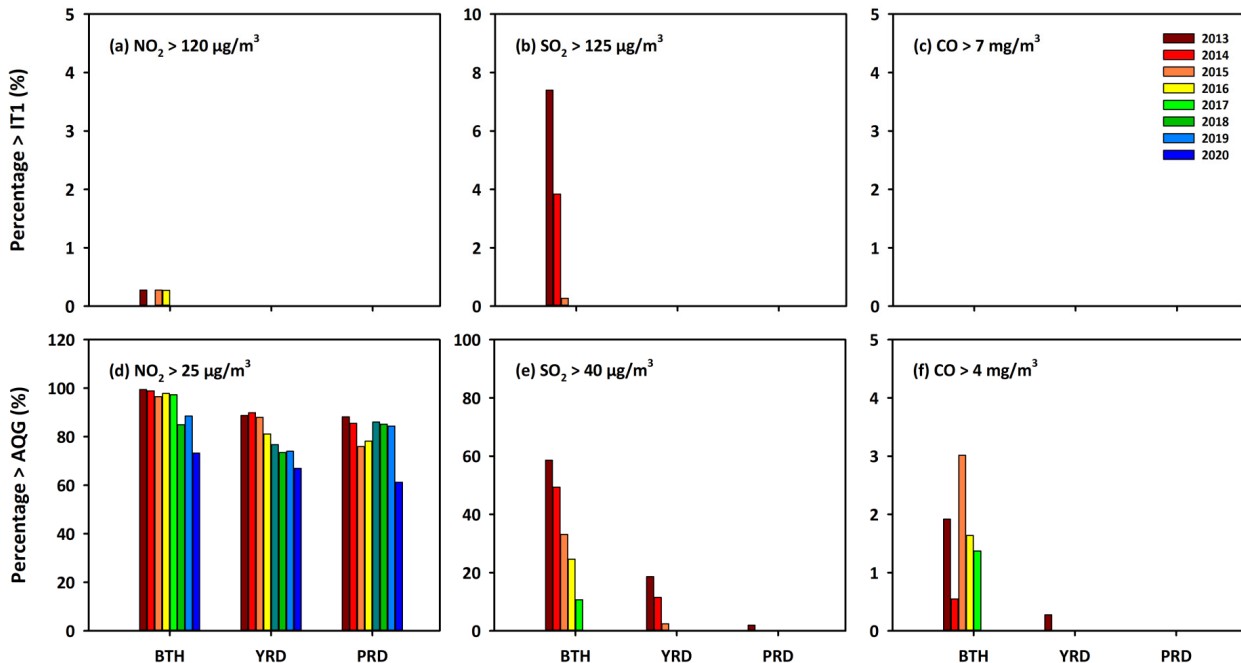

**Figure 9.** Percentage of days (%) exceeding the WHO recommended short-term (a-c) minimum interim target (IT1) and (d-f) desired air quality guidelines (AQG) level for surface $NO_2$, $SO_2$, and CO for each year from 2013 to 2020 in three typical urban agglomerations: the Beijing-Tianjin-Hebei (BTH) region, the Yangtze River Delta (YRD), and the Pearl River Delta (PRD).

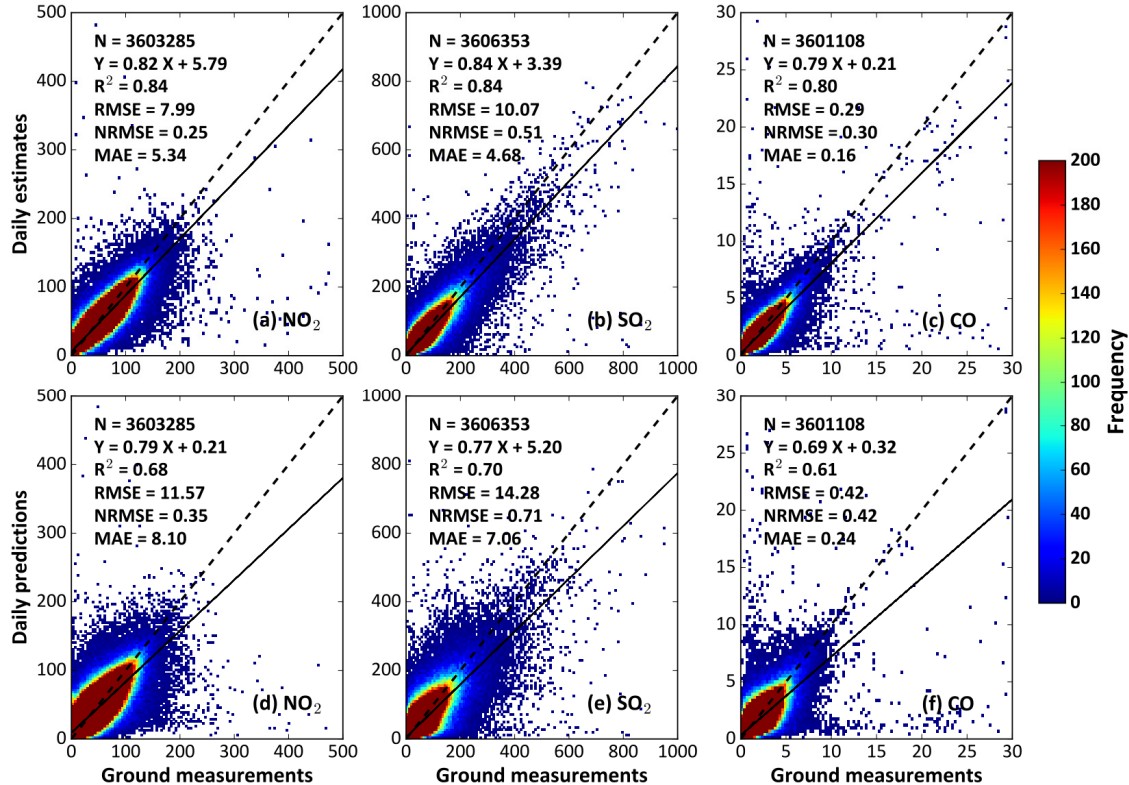

**Figure 10.** Density plots of daily (a-c) estimates and (d-f) predictions of ground-level NO$_2$ (µg/m$^3$), SO$_2$ (µg/m$^3$), and CO (mg/m$^3$) concentrations as a function of ground measurements in China from 2013 to 2020 using the out-of-sample (top panels) and out-of-station (bottom panels) cross-validation methods.

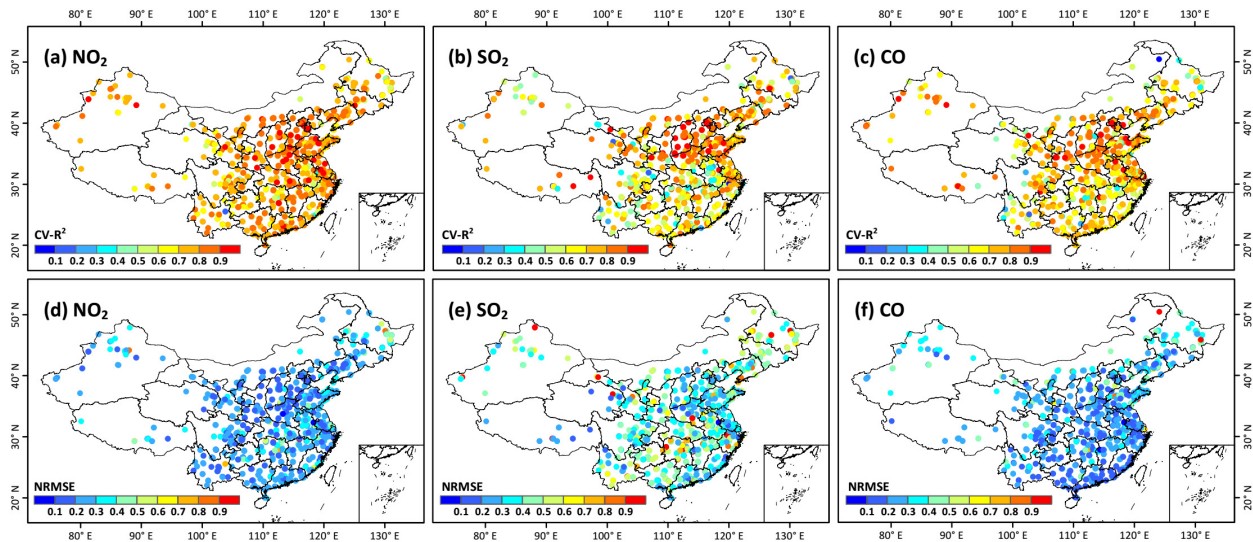

**Figure 11.** Sample-based spatial validation of daily ground-level NO$_2$ (µg/m$^3$), SO$_2$ (µg/m$^3$), and CO (mg/m$^3$) estimates at each individual monitoring station in China from 2013 to 2020: (a-c) accuracy (i.e., CV-R$^2$) and (d-f) uncertainty (i.e., NRMSE).

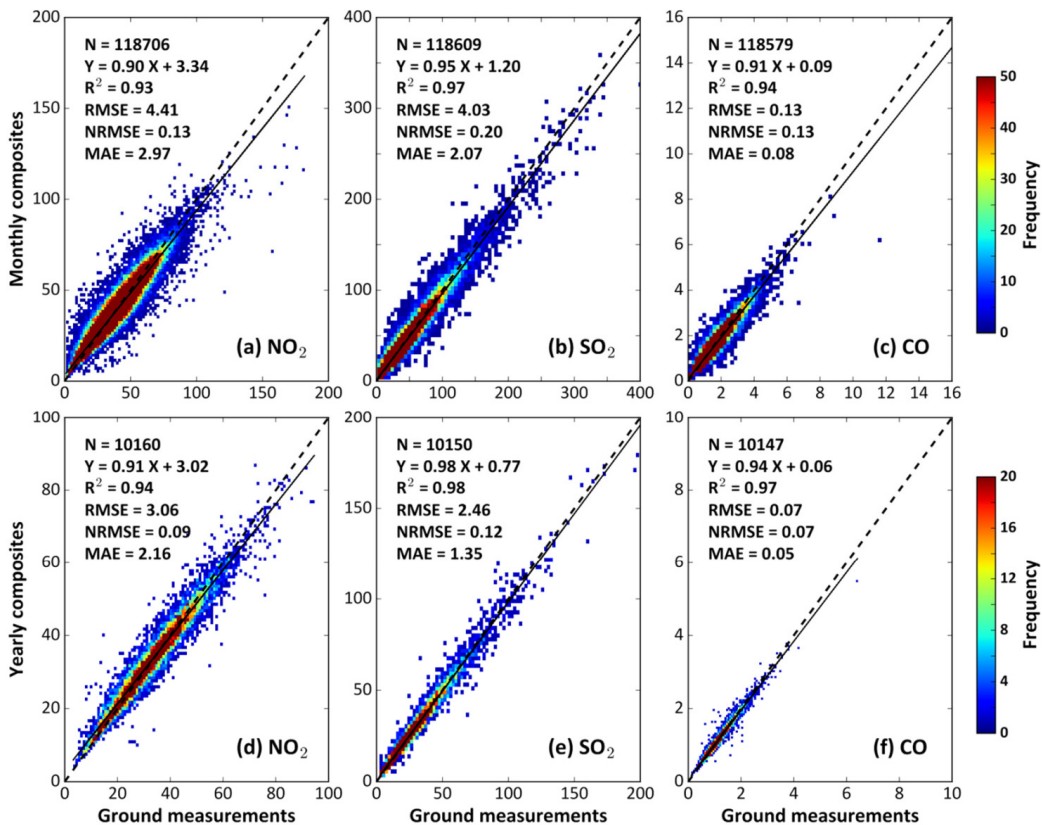

**Figure 12.** Sample-based temporal validation of (a-c) monthly and (d-f) yearly composites of ground-level $NO_2$ ($\mu g/m^3$), $SO_2$ ($\mu g/m^3$), and CO ($mg/m^3$) as a function of ground measurements from 2013 to 2020 in China.

**Tables**

**Table 1.** Statistics of the overall accuracies and predictive abilities of ambient gaseous pollutants for
each year in China from 2013 to 2020.

| Year | Sample size N ($10^3$) | Overall accuracy | | | | | | Predictive ability | | | | | |
|------|------|------|------|------|------|------|------|------|------|------|------|------|------|
| | | $NO_2$ | | $SO_2$ | | CO | | $NO_2$ | | $SO_2$ | | CO | |
| | | $R^2$ | RMSE | $R^2$ | RMSE | $R^2$ | RMSE | $R^2$ | RMSE | $R^2$ | RMSE | $R^2$ | RMSE |
| 2013 | 169 | 0.77 | 12.48 | 0.83 | 17.97 | 0.80 | 0.56 | 0.53 | 18.16 | 0.68 | 25.04 | 0.60 | 0.78 |
| 2014 | 324 | 0.76 | 10.97 | 0.83 | 15.87 | 0.77 | 0.38 | 0.54 | 15.56 | 0.66 | 22.45 | 0.51 | 0.57 |
| 2015 | 518 | 0.79 | 9.34 | 0.80 | 13.71 | 0.74 | 0.38 | 0.61 | 13.10 | 0.61 | 19.49 | 0.50 | 0.55 |
| 2016 | 516 | 0.82 | 8.59 | 0.83 | 11.26 | 0.76 | 0.34 | 0.64 | 12.20 | 0.65 | 16.28 | 0.57 | 0.46 |
| 2017 | 527 | 0.86 | 7.57 | 0.86 | 7.79 | 0.82 | 0.24 | 0.72 | 10.67 | 0.74 | 10.80 | 0.70 | 0.32 |
| 2018 | 513 | 0.87 | 6.92 | 0.83 | 5.61 | 0.82 | 0.20 | 0.76 | 9.33 | 0.68 | 7.80 | 0.69 | 0.26 |
| 2019 | 515 | 0.87 | 6.78 | 0.81 | 4.84 | 0.82 | 0.20 | 0.77 | 9.23 | 0.66 | 6.63 | 0.70 | 0.25 |
| 2020 | 522 | 0.89 | 5.78 | 0.80 | 4.02 | 0.82 | 0.17 | 0.79 | 8.04 | 0.62 | 5.57 | 0.69 | 0.23 |


**Table 2.** Comparison of long-term datasets of different gaseous pollutants in China.

| Species | Model | Missing values | Spatial resolution | Main input | Validation period | CV-R$^2$ | RMSE | Literature |
|---------|-------|----------------|--------------------|------------|-------------------|----------|------|------------|
| NO$_2$ | RF-STK | Yes | 0.25° | OMI | 2013–2016 | 0.62 | 13.3 | (Zhan et al., 2018) |
| | RF-K | Yes | 0.25° | OMI | 2013–2018 | 0.64 | 11.4 | (Dou et al., 2021) |
| | KCS | Yes | 0.125° | OMI | 2014–2016 | 0.72 | 7.9 | (Chen et al., 2019) |
| | LUR | Yes | 0.125° | OMI | 2014–2015 | 0.78 | - | (Xu et al., 2019) |
| | LME | Yes | 0.1° | OMI | 2014–2020 | 0.65 | 7.9 | (Chi et al., 2021) |
| | XGBoost | Yes | 0.125° | TROPOMI | 2018–2020 | 0.67 | 6.4 | (Chi et al., 2022) |
| | XGBoost | Yes | 0.05° | TROPOMI | 2018–2019 | 0.83 | 7.6 | (Liu, 2021) |
| | LightGBM | No | 0.05° | TROPOMI | 2018–2020 | 0.83 | 6.6 | (Wang et al., 2021) |
| | SWDF | No | 0.01° | TROPOMI | 2019–2020 | 0.93 | 4.9 | (Wei et al., 2022b) |
| | STET | No | 0.1° | Big data | 2013–2020 | 0.84 | 8.0 | This study |
| SO$_2$ | RF | No | 0.25° | Emissions | 2013–2014 | 0.64 | 17.1 | (Li et al., 2020) |
| | STET | No | 0.1 | Big data | 2013–2020 | 0.84 | 10.1 | This study |
| CO | RF–STK | Yes | 0.1 | MOPITT | 2013–2016 | 0.51 | 0.54 | (Liu et al., 2019) |
| | LightGBM | No | 0.07° | TROPOMI | 2018–2020 | 0.71 | 0.26 | (Wang et al., 2021) |
| | STET | No | 0.1° | Big data | 2013–2020 | 0.80 | 0.29 | This study |

KCS: kriging-calibrated satellite method; LightGBM: light gradient boosted model; LME: linear mixed effect model;
LUR: land use regression; MOPITT: Measurements of Pollution in the Troposphere; OMI: Ozone Monitoring
Instrument; RF: random forest; RF-K: random forest integrated with K-means; RF-STK: random-forest-spatiotemporal-
kriging model; STET: space-time extremely randomized tree; SWDF: spatiotemporally weighted deep forest;
TROPOMI: TROPOspheric Monitoring Instrument; XGBoost: extreme gradient boosting