# Peer review of "Ground-level gaseous pollutants (NO2, SO2, and CO) in"

_Atmospheric Chemistry and Physics, 2022_

## Author Comment (AC1)

Wei et al. estimated long-term daily seamless different ground-level gaseous pollutants with high accuracy using machine learning and big data by combing monitors, satellites, and models. The public dataset are important to study air quality in China and also have been widely adopted in public health-related studies. The study is well organized and the results are well presented. However, the manuscript still suffers from some flaws. I recommend the manuscript for publication after the following comments are well addressed.

Major comments:
The authors have constructed many air quality dataset (e.g., PM5, PM10) across China. Please introduce the novelty of this study compared with previous studies. I think it is essential to add these contents in the introduction.
Response: Yes, we have constructed a virtually a complete set of major air quality parameters concerning both gaseous and particulate pollutants over a long period of time across China. While we have published a few studies on different parameters, this study adds to the list with the following unique aspects. First, it adds two new species of $SO_2$ and CO for the first time. Instead of devoting to a single pollutant, this paper deals with all gaseous pollutants of compatible quality over the same period with the same spatial coverage and resolution. This is highly valuable for the sake of studying their variations, relative proportions and attribution of emission sources. Per your suggestion, they are clarified in the following revised texts:

"To date, we have combined the advantages of artificial intelligence and big data to construct a virtually complete set of major air quality parameters concerning both particulate and gaseous pollutants over a long period of time across China, including $PM_1$ (2000–Present, Wei et al., 2019), $PM_{2.5}$ (2000–Present, Wei et al., 2020; Wei et al., 2021a), $PM_{10}$ (2000–Present, Wei et al., 2021b), $O_3$ (1979–Present, Wei et al., 2022a; He et al., 2022), and $NO_2$ (2019–Present, Wei et al., 2022b), serving environmental, public health, economy, and other related research. This study is the continuation of our previous studies, which adds two new species of $SO_2$ and CO for the first time and also dates the data records of $NO_2$ back to 2013. Instead of devoting itself to a single pollutant, this paper deals with all gaseous pollutants of compatible quality over the same period with the same spatial coverage and resolution. In particular, considering that there are few public datasets of these three gaseous pollutants with such spatiotemporal coverages focusing on the whole of China, this is highly valuable for the sake of studying their variations, relative proportions, and attribution of emission sources, as well as their diverse and joint effects of different pollutant species on public health."

The authors should discuss the limitations of this paper and prospects for future work in the conclusion. The development of high-resolution dataset might not be the final aim.
Response: We have discussed the limitations of our study and prospects for future work in the revised conclusion as follows:

"Although a lot of new and/or useful data and analyses are presented in this study, they still suffer from some limitations. For example, input variables related to the emission inventory,

modeled simulations, and assimilations still have considerable uncertainties. More influential factors stemming from regional economic and development differences need to be considered in more powerful artificial intelligence models to improve the prediction accuracy of air pollutants. The spatiotemporal resolutions of gaseous pollutants will be further improved by integrating information from polar-orbiting and geostationary satellites to investigate diurnal variations. In a future study, we will also reconstruct data records over the last two decades and investigate their long-term spatiotemporal variations, filling the gap of missing observations. This will help us understand their formation mechanisms and impacts on fine particulate matter and ozone pollution in China."

Specific comments:

Line 41-43: Please spell out these abbreviations, e.g., NOx, VOCs, et al. Also, please double-check and correct such issues throughout the paper.

Response: We have corrected and spelled out all abbreviations throughout the paper.

Lines 48 and 54: Should be MEE and WHO.

Response: Corrected.

Lines 64-69: The authors are suggested to highlight the main purpose and provide more descriptions of the main work here to enrich the Introduction.

Response: We have clarified the main purpose and added more descriptions of our study in the revised Introduction as follows:

"In view of the above problems, the purpose of this paper is to reconstruct daily concentrations of three ambient gaseous pollutants (i.e., $NO_2$, $SO_2$, and CO) in China. To this end, relying on the dense national ground-based observation network and big data, including satellite remote sensing products, meteorological reanalysis, chemical model simulations, and emission inventories, we are capable of mapping three pollutant gases seamlessly (100% spatial coverage) on a daily basis at a uniform spatial resolution of 10 km since 2013 in China. Estimates were made using an extended and powerful machine-learning model incorporating spatiotemporal information, i.e., space-time extra-trees. Natural and anthropogenic effects on air pollution, including their physical mechanisms and chemical reactions, were accounted for in the modeling. Using this dataset, spatiotemporal variations of the gaseous pollutants, the impacts of environmental protection policies and the COVID-19 epidemic, and population risk exposure to gaseous pollution are investigated."

Lines 83-88: A long sentence suggests splitting.

Response: Done per your suggestion.

Line 97: $0.1° \times 0.1°$?

Response: Corrected.

Figures 2 and 3: Please clarify which cross-validated method was used.

Response: We have clarified this in the captions of these figures.

Section 3.2.3: Besides annual variations, it is also interesting to see how three gaseous pollutants changed in different seasons on both the national and regional scales during the study period.

Response: Thanks for your suggestion. We have discussed the changes in gaseous pollutants at the seasonal level in China and three key regions in the revised Section 3.2.2 as follows:

"Large seasonal differences were observed in the amplitude of gaseous pollutant (Figure 6), e.g., surface $NO_2$ decreased the most in winter, especially in the three urban agglomerations (↓24–31%), changing the least in autumn (especially in the YRD). Surface $SO_2$ showed much larger decreases in all seasons, especially during the cold seasons (↓55–81%), due to the implementation of stricter "ultra-low" emission standards (Q. Zhang et al., 2019; Li et al., 2022a). Surface CO had similar seasonal changes as $SO_2$ but 1.5–3.3 times smaller in amplitude."

[Figure]

**Figure 6.** Relative changes (%) in seasonal mean surface $NO_2$, $SO_2$, and CO concentrations between 2013 and 2020 over (a) China, (b) the Beijing-Tianjin-Hebei (BTH) region, (c) the Yangtze River Delta (YRD), and (d) the Pearl River Delta (PRD).

Lines 286 and 294: References are needed to support the evidence here.
Response: Done per your suggestion.

Figures 9 and 10: Since the air quality guidelines have been newly updated in 2021, it is suggested to show the spatial distributions and variations of the percentage of polluted days exceeding both the WHO recommended long-term and short-term AQG levels and interim targets.

Response: Thanks for your suggestions. We have updated this section by discussing the spatiotemporal variations of national and regional polluted days according to the new WHO recommended AQG levels and interim targets in the revision as follows:

"With the daily seamless datasets, we can evaluate the spatial and temporal variations of short-term population-risk exposure to the three gaseous pollutants by calculating the number of days in a given year exceeding the new recommended short-term minimum interim target (IT1) and desired air quality guidelines (AQG) level defined by the WHO in 2021 (WHO, 2021). The area exceeding the recommended levels (i.e., daily $NO_2$ > 120 μg/m$^3$, $SO_2$ > 125 μg/m$^3$, and CO > 7 mg/m$^3$) was generally small in eastern China (Figure S7). High $NO_2$-exposure risks were mainly found in Beijing and Hebei Province and a handful of big cities (e.g., Jinan, Wuhan, Shanghai, and Guangzhou), while high $SO_2$-exposure risks were mainly observed in Hebei, Shandong, and Shaanxi Provinces. The risk of high CO pollution was small, only found in some scattered areas in the NCP. In general, both the area and the possibility of occurrence exposure to high pollution has gradually decreased over time, almost disappearing since 2018.

By contrast, most areas of eastern China had a surface $NO_2$ exposure exceeding the AQG level (Figure 8), especially in the north and economically developed areas in the south (proportion > 80%). Both the extent and intensity are decreasing over time, but it is still a problem, suggesting that stronger $NO_x$ controls are needed in the future. Most of the main air pollution transmission belt in China (i.e., the "2 + 26" cities, Figure 1) had surface $SO_2$ levels exceeding the AQG level at the beginning of the study period. Thanks to strict control measures, these polluted areas sharply decreased after 2015, almost disappearing in 2020. Controlling CO was much more successful in China, with less than 10% of the days in the BTH exceeding the acceptable standard in the early part of the study period. Most areas have reached the CO AQG level since 2018.

[Figure]

**Figure 8.** Spatial distributions of the percentage of days exceeding the WHO recommended short-term desired air quality guidelines level for surface $NO_2$ (daily mean > 25 μg/m$^3$), $SO_2$

(daily mean > 40 μg/m$^3$), and CO (daily mean > 4 mg/m$^3$) for each year from 2013 to 2020 in populated areas in eastern China.

Figure 9 shows the percentage of days with pollution levels exceeding WHO air quality standards in three key regions. BTH was the only region experiencing high NO$_2$ and SO$_2$ exposure risks (i.e., daily mean > IT1), dropping to zero since 2017 and 2016, while YRD and PRD had no high risks of exposure to the three gaseous pollutants (Figure 9a-b). There was also no regional high CO-pollution risk (Figure 9c). However, although declining continuously, regional surface NO$_2$ levels failed to meet the short-term AQG level in 2020, with 61–73% of the days exceeding the AQG level. More efforts toward mitigating NO$_2$ levels in these key regions are thus needed. Continual decreases in the number of days above the AQG level were also observed in surface SO$_2$, reducing to near zero in 2014, 2016, and 2018 in the PRD, YRD, and BTH, respectively. Less than 3% of the days in the BTH and YRD had surface CO levels exceeding the AQG level. Surface CO levels were always below the AQG level in the PRD."

[Figure]

**Figure 9.** Percentage of days (%) exceeding the WHO recommended short-term (a-c) minimum interim target (IT1) and (d-f) desired air quality guidelines (AQG) level for surface NO$_2$, SO$_2$, and CO for each year from 2013 to 2020 in three typical urban agglomerations: the Beijing-Tianjin-Hebei (BTH) region, the Yangtze River Delta (YRD), and the Pearl River Delta (PRD)."

---

## Author Comment (AC2)

The manuscript by Wei and colleagues titled "Ground-level gaseous pollutants across China: daily seamless mapping and long-term spatiotemporal variations" professes to generate seamless daily maps of three major pollutant gases, NO2, SO2, and CO, across China from 2013 to 2020 at a uniform spatial resolution of 10 km. While the topic is overall still quite interesting for the global air quality community, the manuscript has a number of serious scientific flaws which unfortunately led me to the recommendation of rejection. These issues are explained below, but are also clearly noted and commented upon in the annotated text that is included with this review.

Response: We appreciate the time and effort the reviewer spent on this manuscript and the insightful comments and constructive suggestions. We have carefully considered each comment and addressed them one by one in the revised manuscript. Responses to each comment made as annotated text in the manuscript are also given

The main premise of the generation of the daily maps of gaseous concentration is that the authors used artificial intelligence technologies and big data to produce these maps. The model used is not at all adequately described: it is simply named, Space-Time Extra-Tree, and a reference to a previous work that produced O3 maps is given. This is not at all sufficient for the reader of this work to assess the model, its strengths, its limitations, nor to assess whether a model that functioned well for one gas would work for another gas.

Response: Thanks for your suggestion. We have added more descriptions of the extended model, including its strengths and limitations, in the revision as follows:

"Here, the developed Space-Time Extra-Tree (STET) model, integrating spatiotemporal autocorrelations of and differences in air pollutants to the Extremely Randomized Trees (ERT) (Wei et al., 2022a), was extended to estimate surface gaseous pollutants, i.e., $NO_2$, $SO_2$, and CO. ERT is an ensemble machine-learning model based on the decision tree, capable of solving the nonparametric multivariable nonlinear regression problem. Ensemble learning can avoid the lack of learning ability of a single learner, greatly improving accuracy. The introduced randomness enhances the model's anti-noise ability and minimizes the sensitivity to outliers and multicollinearity issues. It can handle high latitude, discrete or continuous data without data normalization and is easy to implement and parallel. However, several limitations exist, e.g., it is difficult to make predictions beyond the range of training data, and there will be an over-fitting issue on some regression problems with high noise. The training efficiency diminishes with increasing memory occupation when the number of decision trees is large (Geurts et al., 2006).

Compared with traditional tree-based models (e.g., random forest), ERT has a stronger randomness which randomly selects a feature subset at each node split and randomly obtains the optimal branch attributes and thresholds. This helps to create more independent decision trees, further reducing model variance and improving training accuracy (Geurts et al., 2006). The STET model has been successfully applied in estimating high-quality surface $O_3$ in our previous study (Wei et al., 2022a). It is thus extended here to regress the nonlinear conversion relationships between ground-based measurements and the main predictors and auxiliary

factors for other species of gaseous pollutants. For surface NO$_2$, the STET model was applied to the main variables of the satellite tropospheric NO$_2$ column, modelled surface NO$_2$ mass, and NO$_x$ emissions, together with ancillary variables of the previously mentioned meteorological, surface, and population variables (Equation 1). For surface SO$_2$ (Equation 2) and CO (Equation 3), modelled surface SO$_2$ and CO concentrations and SO$_2$ and CO emissions were used as main predictors along with the same auxiliary variables as NO$_2$ to construct the STET models separately."

Section 2.2 is extremely poor in reproducible content in that respect. The input parameters used in the model are not at all adequately described: in section 2.1.2 a long list of satellite, reanalysis, and model datasets are more or less simply named, without the most pertinent details of provenance, usability, references, validation and quality assurance being provided. Exactly how these input parameters were used in the STET model are not explained at all. Furthermore, these datasets have obvious important differences, for e.g. the OMI/GOME2 VCDs and the CAMS reanalysis VCDs, there is no discussion how these were merged into a usable dataset.

Response: We have rewritten this section by introducing each input parameter in detail separately, including their provenance, usability, references, validation and quality assurance, and their role in the model. We have also discussed how to merge the different tropospheric NO$_2$ VCDs and cited the corresponding references in the revision as follows:

**"2.1.2 Main predictors**
A new daily tropospheric NO$_2$ dataset at a horizontal resolution of 0.25° × 0.25° in China (https://doi.org/10.6084/m9.figshare.13126847) was employed, created by Q. He et al. (2020) using a developed framework integrating OMI/Aura Quality Assurance for Essential Climate Variables (QA4ECV) and Global Ozone Monitoring Experiment–2B (GOME-2B) offline tropospheric NO$_2$ retrievals passing quality controls (i.e., cloud fraction < 0.3, surface albedo < 0.3, and solar zenith angle < 85°). The reconstructed tropospheric NO$_2$ agreed well (R = 0.75–0.85) with Multi-AXis Differential Optical Absorption Spectroscopy (MAX-DOAS) measurements (H. He et al., 2020). Through this data fusion, the daily spatial coverage of satellite tropospheric NO$_2$ was significantly improved in China (average = 87%). Areas with a small number of missing values were imputed via a nonparametric machine-learning model by regressing the conversion relationship with Copernicus Atmosphere Monitoring Service (CAMS) tropospheric NO$_2$ assimilations (0.75° × 0.75°), making sure that the interpolation was consistent with the OMI/Aura overpass time (Inness et al., 2019; Y. Wang et al., 2020). The gap-filled tropospheric NO$_2$ was reliable compared with measurements (R = 0.94–0.98) (Wei et al., 2022b). The above two-step gap-filling procedures allowed us to generate a daily seamless tropospheric NO$_2$ dataset that removes the effects of clouds from satellite observations.
Here, the reconstructed daily seamless tropospheric NO$_2$, together with CAMS daily ground-level NO$_2$ assimilations (0.75° × 0.75°) averaged from all 3-hourly data in a day and monthly NO$_x$ anthropogenic emissions (0.1° × 0.1°) (Inness et al., 2019), were used as the main predictors for estimating surface NO$_2$. Limited by the quality of direct satellite observations, daily model-simulated SO$_2$ and CO surface mass concentrations, averaged from all available

data in a day provided by one-hourly Modern-Era Retrospective Analysis for Research and Applications, version 2 (MERRA-2, 0.625° × 0.5°), 3-hourly CAMS (0.75° × 0.75°), and 3-hourly Goddard Earth Observing System Forward-Processing (0.3125° × 0.25°) global reanalyses were used as main predictors to retrieve surface $SO_2$ and CO, together with CAMS monthly $SO_2$ and CO anthropogenic emissions.

**2.1.3 Auxiliary factors**

Meteorological factors have important diverse effects on air pollutants (J. He et al., 2017; R. Li et al., 2019), e.g., the boundary-layer height reflects their vertical distribution and variations (Z. Li et al., 2017; Seo et al., 2017); temperature, humidity, and pressure can affect their photochemical reactions (W. Y. Xu et al., 2011; T. Li et al., 2019; C. Zhang et al., 2019a); and rainfall and wind can also influence their removal, accumulation, and transport (Dickerson et al., 2007; R. Li et al., 2019). Eight daily meteorological variables, provided by the ERA5-Land (0.1° × 0.1°; Muñoz-Sabater et al., 2021) and ERA5 global reanalysis (0.25° × 0.25°; Hersbach et al., 2020), were calculated (i.e., accumulated for precipitation and evaporation while averaged for the others) from all hourly data in a day, used as auxiliary variables to improve the modelling of gaseous pollutants. Other auxiliary remote-sensing data used to describe land-use cover/change [i.e., Moderate Resolution Imaging Spectroradiometer (MODIS) normalized difference vegetation index (NDVI), 0.05° × 0.05°] and population distribution density (i.e., LandScan$^{TM}$, 1 km) were employed as inputs to the machine-learning model because they are highly related to the type of pollutant emission and amounts of anthropogenic emissions, as well as the surface terrain [i.e., Shuttle Radar Topography Mission (SRTM) digital elevation model (DEM), 90m], which can affect the transmission of air pollutants. Table 1 provides detailed information about all the data used in this study."

The meteorological ERA5 data are on a 3h level, how were these turned into daily means, and what does it actually mean that they did, etc., is an issue also not discussed.
Response: Meteorological ERA5 data are hourly. Here, for precipitation and evaporation, the hourly amounts on a day are accumulated to obtain the daily values. For the other meteorological variables, all hourly data are averaged to obtain daily values. These auxiliary variables were used to improve the modelling of gaseous pollutants. We have clarified this in the revision (see the response to your last comment).

The main input parameters, both for the training of the model and the verification of the model, i.e. the ground-based measurements are not at all adequately described. In section 2.1.1 it is not at all clear what these "reference-grade ground-based monitoring" stations are, how they were chosen, if and how the data pass QA/QC protocols, what the reference state is, how these stations were split for the verification of the STET and the training of the STET, how the gaps in the datasets were dealt with, how the hourly observations were turned into daily, etc.
Response: We have described the ground-based measurements used in detail and clarified how the ground monitors were used for training and validation in the revision:

**"2.1.1 Ground-based measurements**

Hourly measurements of ground-level $NO_2$, $SO_2$, and CO concentrations from ~2000 reference-grade ground-based monitoring stations (Figure 1) collected from the China National Environmental Monitoring Centre (CNEMC) network (https://www.cnemc.cn/en/) were employed in the study. This network includes urban assessing stations, regional assessing stations, background stations, source impact stations, and traffic stations, set up in a reasonable overall layout that covers industrial (~14%), urban (~31%), suburban (~39%), and rural (~16%) areas to improve the spatial representation, continuity, and comparability of observations (HJ 664-2013) (MEE, 2013a). $NO_2$ is measured by chemiluminescence and differential optical absorption spectroscopy (DOAS), and $SO_2$ uses ultraviolet fluorescence and DOAS, while CO adopts non-dispersive infrared spectroscopy and gas filter correlation infrared spectroscopy. These measurements have been fully validated and have the same average error of indication of ±2% F.S. for the three gaseous pollutants considered here, with additional quality-control checks such as zero and span noise and zero and span drift (HJ 193-2013 and HJ 654-2013) (MEE, 2013b, 2013c). They have also been used as ground truth in almost all air pollutant modelling studies in China (Ma et al., 2022; B. Zhang et al., 2022a). All stations use the same technique to measure each gas routinely and continuously 24 hours a day at about the sea level without time series gaps. However, the reference state (i.e., observational conditions like temperature and pressure) changed from the standard condition (i.e., 273 K and 1013 hPa) to the room condition (i.e., 298 K and 1013 hPa) on 31 August 2018 (MEE, 2018a). We thus first converted observations of the three gaseous pollutants after this date to the uniform standard condition for consistency. Here, daily values for each air pollutant were averaged from at least 30% of valid hourly measurements at each station in each year from 2013 to 2020."

**"3.3 Data quality assessment**
An additional out-of-station 10-CV approach was used to validate the prediction accuracy of gaseous pollutants, performed based on measurements from ground monitoring stations. These measurements were randomly divided into ten subsets, of which data samples from nine subsets were used for model training and the remaining subset for model validation. This was done 10 times, in turn, to ensure that data from all stations were tested. This procedure generates independent training samples and test samples made in different locations, used to indicate the spatial prediction ability of the model in areas where ground-based measurements are unavailable (S. Wu et al., 2021; Wei et al., 2022a)."

The results are not sufficient to support the interpretations and conclusions. The section starts, not with the expected maps of the input parameters, maps of the output parameters and maps of the ground-based stations, but with model performance scatter plots which are not at all explained as to what is being compared to what. Absolute levels are also provided for biases which have no meaning whatsoever if the actual levels of these gases around China are not provided to begin with.
Response: We apologize for the confusing logic in describing the results. We have adjusted this part by first introducing the seamless mapping results in sequence (i.e., daily and seasonal distributions, temporal changes like COVID lockdown effects, and population-risk exposure to gaseous pollutants). We then evaluated the quality of the datasets by comparing them with

ground measurements in the revision, as suggested. Note that we have replaced RMSE absolute values with normalized RMSE (NRMSE) values to better describe the estimated biases and uncertainties for gaseous pollutants in China in the revision.

A section is also provided, 3.3, where this dataset is being compared, basically via Table 4, to numerous other related works. How the comparisons were made is unclear, how the statistics shown in the table were created is unclear, how so different datasets were homoegenized before comparison is unclear, and the final statement that our gaseous pollutant datasets are superior to those from the studies is not at all shown in this work. It is impossible to assess the interpretations and conclusions stated by the authors based on the information provided in the results section.

Response: Yes, in this section, we compared our results with those from previous studies on the estimation of the three gaseous pollutants using different developed models focusing on the whole of China. Here, only those studies applying the same out-of-sample cross-validation approach against ground-based measurements collected from the same CNEMC network were selected. The statistics shown in the table come from the publications themselves because their generated datasets are not publicly available. We have applied the same validation method and ground measurements as those used in the previous studies. We have clarified this in the revision.

Another premise that the authors mention numerous times, in the title even, is that the new dataset is long-term and that it will benefit future (especially short-term) air pollution and environmental health-related studies. They provide a section, 3.4, where they enumerate successful applications however it is unclear if these studies used their previous work on O3, or other similar works. The benefits of this work should be clearly stated, to support this work, and not generalities.

Response: Results from all the studies listed in this section were obtained using the three gaseous pollutant (i.e., $NO_2$, $SO_2$, and CO) datasets generated in this study. We have clearly stated the benefits according to your suggestion in the revision as follows:

"A large number of studies have used the three gaseous pollutant datasets generated in this study to study their single or joint impacts on environmental health from both long-term and short-term perspectives, benefiting from the unique daily spatially seamless coverage. For example, a nearly linear relationship between long-term ambient $NO_2$ and adult mortality in China was observed (Y. Zhang et al., 2022). Y. Wang et al. (2023) reported that ambient $NO_2$ hindered the survival of middle-aged and elderly people. Long-term $SO_2$ and CO exposure can increase the incidence rate of visual impairment in children in China (L. Chen et al., 2022a), and short-term exposure to ambient CO can significantly increase the probability of hospitalization for stroke sequelae (R. Wang et al., 2022). Regional and national cohort studies have shown that exposure, especially short-term exposure, to multiple ambient gaseous ($NO_2$, $SO_2$, and CO) and particulate pollutants have negative effects of varying degrees on a variety of diseases, like cause-specific cardiovascular disease (R. Xu et al., 2022a,b), ischemic and hemorrhagic stroke ( Cai et al., 2022; He et al., 2022; H. Wu et al., 2022b; R. Xu et al., 2022c), asthma mortality (W. Liu et al., 2022), dementia mortality (T. Liu

et al., 2022), metabolic syndrome (S. Han et al., 2022), blood pressure (Song et al., 2022; H. Wu et al., 2022a), renal function (S. Li et al., 2022), neurodevelopmental delay (X. Su et al., 2022), serum liver enzymes (Y. Li et al., 2022), overweight and obesity (L. Chen et al., 2022b), insomnia (J. Xu et al., 2021), and sleep quality (L. Wang et al., 2022)."

Concluding, while is it possible that this work has potential for air quality-related studies, through the current manuscript the description of experiments and calculations is not sufficiently complete and precise to allow their reproduction by fellow scientists and provide traceability of results. I recommend to the authors to take the opportunity of this review to reconsider their strategy for their future publications.

Response: We appreciate your comments and suggestions, which have greatly improved our paper. We have added more descriptions of the experiments and calculations according to your suggestions, making this study reproducible by others. We will keep this in mind as we work on our future publications.

**Response to each comment copied from the annotated text in the paper.**
Line 19: You are showing eight years in your work. This cannot be considered a "long period". I am weary about not only the main take away message of this work but also the usability by the community.

Response: We have deleted "long period" from the sentence and removed such descriptions throughout the paper.

Line 21: Acronyms need to be added.

Response: Done per your suggestion.

Line 22: Cross-validation between what and what? this information should be clear at this stage of the abstract.

Response: We have added "between our estimates and ground observations" here.

Line 23: "out-of-bag". Please consult with the journal if this terminology is encouraged.

Response: We have deleted it here.

Lines 24-26: This is too vague as a phrase for an abstract.

Response: We have rephrased this sentence as "We found that the COVID-19 lockdown had sustained impacts on gaseous pollutants, where surface CO recovered to its normal level in China on around the 34th day after the Lunar New Year, while surface $SO_2$ and $NO_2$ rebounded more than twice slower due to more CO emissions from increased residents' indoor cooking and atmospheric oxidation capacity."

Lines 26-27: In absolute numbers, this really does not mean much. Only relative terms. Plus the std is needed as well, especially for values calculated for 7 years only.

Response: We have rephrased this sentence as "Surface $NO_2$, $SO_2$, and CO reached their peak annual concentrations of $21.3 \pm 8.8$ μg/m$^3$, $23.1 \pm 13.3$ μg/m$^3$, and $1.01 \pm 0.29$ mg/m$^3$ in 2013, then continuously declined over time by 12%, 55%, and 17%, respectively, until 2020."

Line 28: "three urban agglomerations" Which is where? what is the source of CO, SO2 & NO2 for all three?
Response: We have deleted this from the abstract.

Lines 28-32: Too vague for an abstract. What is the reason for this?
Response: We have clarified the reasons here and rephrased the sentence as "The declining rates were more prominent from 2013 to 2017 due to the sharper reductions in anthropogenic emissions but have slowed down in recent years. Nevertheless, people still suffer from high-frequency risk exposure to surface $NO_2$ in eastern China, while surface $SO_2$ and CO have almost reached the recommended air quality guidelines level since 2018, benefiting from the implemented stricter "ultra-low" emission standards."

Lines 32-33: "ChinaHighNO2, ChinaHighSO2, and ChinaHighCO" What do these names mean?
Response: They are the names of the three gaseous pollutant datasets produced in this study. This has been deleted from the abstract.

Line 38: Li et al., 2017a should precede 2017b.
Response: In this paper, the following way of citing other work was followed. To distinguish between first authors with the same surname, their first-name initials were used when citing their work. This avoids using suffixes "a", "b", etc. after the publication year if the publication year happens to be the same. Suffixes "a", "b", etc. were only used if referring to works published in the same year by the same first author.

Line 44: followings -> following; matters -> matter
Response: Corrected.

Line 52: "By contrast, ground-level gaseous pollutants have been much less studied." A quick search in Scopus using simply air quality and NOx and China seems to disrepute this fact.
Response: We have removed this statement from the revision.

Line 55: Add acronyms and spacecraft.
Response: Done per your suggestion.

Line 64: "a long-term". 7 years is not long term. Re-phrase.
Response: We have deleted it from the sentence.

Lines 74-76: This is really poor in describing the datasets. Where are these stations? which network do they belong to? are they open source? have they been validated? which method is used to measure the gases? have they been already used in other studies? what percentage of them are urban, suburban, industrial, rural? are they all at sea level, or are some above the PBL? All this information has to be presented here.
Response: We have added a figure showing the locations of the stations used in this study.

They belong to the China National Environmental Monitoring Centre (CNEMC) network (open-source available at https://www.cnemc.cn/en/). This network includes urban assessing stations, regional assessing stations, background stations, source impact stations, and traffic stations, set up in a reasonable overall layout that covers industrial (~14%), urban (~31%), suburban (~39%), and rural (~16%) areas to improve the spatial representations, continuity, and comparability of observations (HJ 664-2013) (MEE, 2013a). $NO_2$ is measured by chemiluminescence and differential optical absorption spectroscopy (DOAS), and $SO_2$ uses ultraviolet fluorescence and DOAS, while CO adopts non-dispersive infrared spectroscopy and gas filter correlation infrared spectroscopy. All stations use the same technique to measure each gas routinely and continuously 24 hours a day at about the sea level without time series gaps. These measurements have been fully validated and have the same average error of indication of ±2% F.S. for the three gaseous pollutants considered here (HJ 193-2013 and HJ 654-2013) (MEE, 2013b, 2013c). They have been used as ground truth in almost all air pollutant modelling studies in China (Ma et al., 2022; B. Zhang et al., 2022). We have clarified this in the revised Section 2.1.1.

Line 76: What is this reference state? why do you need to convert the concentrations? what is the conversion about, i.e., from what to what?
Response: The reference state refers to observation conditions like the temperature and pressure of gaseous pollutants. The reference state changed from the standard condition (i.e., 273 K and 1013 hPa) to the room condition (i.e., 298 K and 1013 hPa) on 31 August 2018 (MEE, 2018a). We thus first converted observations of the three gaseous pollutants after this date to the uniform standard condition for consistency. We have clarified this in the revision.

Lines 78-79: How many hourly measurements did you permit in order to make the daily mean? did you simply average or is there an error estimate? are all 2000 stations using the same technique to measure each gas? did they all have timeseries without gaps?
Response: Here, daily values for each air pollutant were averaged from at least 30% of valid hourly measurements at each station in each year. All stations use the same technique to measure each gas routinely and continuously 24 hours a day at about the sea level without time series gaps. We have clarified this in the revision.

Line 79: Who is providing this QA/QC? what does it entail? you need references and a discussion showing the quality of the input data.
Response: The QA/QC is performed by the China National Environmental Monitoring Centre (CNEMC) during specification and test procedures for continuous automated monitoring of gaseous pollutants, including multiple quality-control checks like zero and span noise and zero and span drift (HJ 193-2013 and HJ 654-2013) (MEE, 2013b, 2013c). We have clarified this in the revision.

Line 84: Which version of the data? where did you download them from? what QA/QC did you apply? for which geographical region? add proper references to the people who created the data, add validation paper of this data.
Response: This new daily tropospheric $NO_2$ dataset focused on China was created by He et al.

(2020) using a developed framework integrating OMI/Aura Quality Assurance for Essential Climate Variables (QA4ECV) and Global Ozone Monitoring Experiment–2B (GOME-2B) offline tropospheric $NO_2$ retrievals passing quality controls (i.e., cloud fraction < 0.3, surface albedo < 0.3, and solar zenith angle < 85°). The reconstructed tropospheric $NO_2$ agrees well (R = 0.75–0.85) with Multi-AXis Differential Optical Absorption Spectroscopy (MAX-DOAS) measurements. This dataset has only one version (so no version ID) and can be downloaded directly from https://doi.org/10.6084/m9.figshare.13126847. We have added information about this and cited the relevant reference for this dataset in the revision as you suggested. Details can be found in He et al. (2020):

He, Q., Qin, K., Cohen, J. B., Loyola, D., Li, D., Shi, J., and Xue, Y.: Spatially and temporally coherent reconstruction of tropospheric $NO_2$ over China combining OMI and GOME-2B measurements, Environmental Research Letters, 15, 125011, https://doi.org/10.1088/1748-9326/abc7df, 2020.

Furthermore, it is impossible to have daily seamless tropospheric NO2 data from satellite observations due to clouds. How did you tackle that? Which CAMS simulations? how did you gap-fill? how did you deal with the jumps in absolute values? References are needed for the CAMS simulations and their validation.

Response: Daily seamless tropospheric $NO_2$ data were generated via a two-step gap-filling procedure. The first step is data fusion by integrating OMI/Aura and GOME-2B tropospheric $NO_2$ retrievals. The next step is data imputation by regressing CAMS tropospheric $NO_2$ assimilations (Inness et al., 2019) with a machine-learning model. We have clarified this and cited the reference for the CAMS simulations and validation in the revision.

"Through this data fusion, the daily spatial coverage of satellite tropospheric $NO_2$ was significantly improved in China (average = 87%). Areas with a small number of missing values were imputed via a nonparametric machine-learning model by regressing the conversion relationship with Copernicus Atmosphere Monitoring Service (CAMS) tropospheric $NO_2$ assimilations (0.75° × 0.75°), making sure that the interpolation was consistent with the OMI/Aura overpass time (Inness et al., 2019; Y. Wang et al., 2020). The gap-filled tropospheric $NO_2$ was reliable compared with measurements (R = 0.94–0.98) (Wei et al., 2022b). The above two-step gap-filling procedures allowed us to generate a daily seamless tropospheric $NO_2$ dataset that removes the effects of clouds from satellite observations."

Inness, A., Ades, M., Agustí-Panareda, A., Barré, J., Benedictow, A., Blechschmidt, A. M., Dominguez, J. J., Engelen, R., Eskes, H., Flemming, J., Huijnen, V., Jones, L., Kipling, Z., Massart, S., Parrington, M., Peuch, V. H., Razinger, M., Remy, S., Schulz, M., and Suttie, M.: The CAMS reanalysis of atmospheric composition, Atmos. Chem. Phys., 19, 3515-3556, 10.5194/acp-19-3515-2019, 2019.

Wang, Y., Ma, Y. F., Eskes, H., Inness, A., Flemming, J., and Brasseur, G. P.: Evaluation of the CAMS global atmospheric trace gas reanalysis 2003–2016 using aircraft campaign observations, Atmos. Chem. Phys., 20, 4493-4521, 10.5194/acp-20-4493-2020, 2020.

Lines 86-88: What did you use NDVI data for NO2 studies? why annual population? These phrases make no sense one after the other. Simply enumerating datasets is not a proper description either of the datasets nor as to how they were used.

Response: Thanks for your suggestion. We have introduced and described each dataset separately in the revision. NDVI and population are two auxiliary remote-sensing variables input to the machine-learning model, used to describe land-use cover/change and population distribution density because they are highly related to the type of pollutant emission and amounts of anthropogenic emissions. Annual population data is used here because the population does not change much in a year. Note that LandScan$^{TM}$ population information is widely used, with high spatial (1 km) and temporal (updated annually) resolutions.

Lines 88-89: Why do you need meteorological fields? which versions of the reanalysis did you use?

Response: The reason is that meteorological factors have important diverse effects on air pollutants (J. He et al., 2017; R. Li et al., 2019), e.g., the boundary-layer height reflects their vertical distribution and variations (Z. Li et al., 2017; Seo et al., 2017); temperature, humidity, and pressure can affect their photochemical reactions (W. Y. Xu et al., 2011; T. Li et al., 2019; C. Zhang et al., 2019); and rainfall and wind can also influence their removal, accumulation, and transport (Dickerson et al., 2007; R. Li et al., 2019). Meteorological fields are thus used as auxiliary variables to improve the modelling of gaseous pollutants and are provided by the ERA5 global reanalysis. ERA5 has only one version (so no version ID) and can be downloaded directly from the Climate Data Store (https://cds.climate.copernicus.eu/). Table S1 now includes an extra column showing data versions (where applicable).

It is also muddled, CAMS is mentioned in line 85 and the again in line 95.

Response: They are two different things. The former are CAMS tropospheric $NO_2$ assimilations, used for filling the gaps in satellite tropospheric $NO_2$, while the latter are CAMS daily ground-level $NO_2$ assimilations, used as one of the main predictors for estimating surface $NO_2$. We have clarified this in the revision.

This paragraph is extremely poor, it does not described datasets properly nor does it explain what these datasets will be used for.

Response: We have rephrased this part by describing each dataset separately and explaining their purposes according to your suggestion as follows:

**"2.1.1 Main predictors**
A new daily tropospheric $NO_2$ dataset at a horizontal resolution of 0.25° × 0.25° in China (https://doi.org/10.6084/m9.figshare.13126847) was employed, created by Q. He et al. (2020) using a developed framework integrating OMI/Aura Quality Assurance for Essential Climate Variables (QA4ECV) and Global Ozone Monitoring Experiment–2B (GOME-2B) offline tropospheric $NO_2$ retrievals passing quality controls (i.e., cloud fraction < 0.3, surface albedo < 0.3, and solar zenith angle < 85°). The reconstructed tropospheric $NO_2$ agreed well (R = 0.75–0.85) with Multi-AXis Differential Optical Absorption Spectroscopy (MAX-DOAS)

measurements (H. He et al., 2020). Through this data fusion, the daily spatial coverage of satellite tropospheric $NO_2$ was significantly improved in China (average = 87%). Areas with a small number of missing values were imputed via a nonparametric machine-learning model by regressing the conversion relationship with Copernicus Atmosphere Monitoring Service (CAMS) tropospheric $NO_2$ assimilations (0.75° × 0.75°), making sure that the interpolation was consistent with the OMI/Aura overpass time (Inness et al., 2019; Y. Wang et al., 2020). The gap-filled tropospheric $NO_2$ was reliable compared with measurements (R = 0.94–0.98) (Wei et al., 2022b). The above two-step gap-filling procedures allowed us to generate a daily seamless tropospheric $NO_2$ dataset that removes the effects of clouds from satellite observations.

Here, the reconstructed daily seamless tropospheric $NO_2$, together with CAMS daily ground-level $NO_2$ assimilations (0.75° × 0.75°) averaged from all 3-hourly data in a day and monthly $NO_x$ anthropogenic emissions (0.1° × 0.1°) (Inness et al., 2019), were used as the main predictors for estimating surface $NO_2$. Limited by the quality of direct satellite observations, daily model-simulated $SO_2$ and CO surface mass concentrations, averaged from all available data in a day provided by one-hourly Modern-Era Retrospective Analysis for Research and Applications, version 2 (MERRA-2, 0.625° × 0.5°), 3-hourly CAMS (0.75° × 0.75°), and 3-hourly Goddard Earth Observing System Forward-Processing (0.3125° × 0.25°) global reanalyses were used as main predictors to retrieve surface $SO_2$ and CO, together with CAMS monthly $SO_2$ and CO anthropogenic emissions.

**2.1.2 Auxiliary factors**

Meteorological factors have important diverse effects on air pollutants (J. He et al., 2017; R. Li et al., 2019), e.g., the boundary-layer height reflects their vertical distribution and variations (Z. Li et al., 2017; Seo et al., 2017); temperature, humidity, and pressure can affect their photochemical reactions (W. Y. Xu et al., 2011; T. Li et al., 2019; C. Zhang et al., 2019a); and rainfall and wind can also influence their removal, accumulation, and transport (Dickerson et al., 2007; R. Li et al., 2019). Eight daily meteorological variables, provided by the ERA5-Land (0.1° × 0.1°; Muñoz-Sabater et al., 2021) and ERA5 global reanalysis (0.25° × 0.25°; Hersbach et al., 2020), were calculated (i.e., accumulated for precipitation and evaporation while averaged for the others) from all hourly data in a day, used as auxiliary variables to improve the modelling of gaseous pollutants. Other auxiliary remote-sensing data used to describe land-use cover/change [i.e., Moderate Resolution Imaging Spectroradiometer (MODIS) normalized difference vegetation index (NDVI), 0.05° × 0.05°] and population distribution density (i.e., LandScan$^{TM}$, 1 km) were employed as inputs to the machine-learning model because they are highly related to the type of pollutant emission and amounts of anthropogenic emissions, as well as the surface terrain [i.e., Shuttle Radar Topography Mission (SRTM) digital elevation model (DEM), 90m], which can affect the transmission of air pollutants. Table 1 provides detailed information about all the data used in this study."

Line 113-114: You have not mentioned using satellite SO2 and CO. This phrase is not understood.
Response: We have deleted this redundant sentence to avoid ambiguity.

Equations 1-3: These equations make not sense to whoever does not know what fstet does. You need to described the methodology in much more detail, a simple reference to a previous paper is not enough, nor does it convince the reader that the methodology will work for the species discussed here.

Response: We have described the methodology and extended model in more detail, including strengths and limitations, in the revision as follows:

"Here, the developed Space-Time Extra-Tree (STET) model, integrating spatiotemporal autocorrelations of and differences in air pollutants to the Extremely Randomized Trees (ERT) (Wei et al., 2022a), was extended to estimate other surface pollutant gases, i.e., $NO_2$, $SO_2$, and CO. ERT is an ensemble machine-learning model based on the decision tree, capable of solving the nonparametric multivariable nonlinear regression problem. Ensemble learning can avoid the lack of learning ability of a single learner, greatly improving accuracy. The introduced randomness enhances the model's anti-noise ability and minimizes the sensitivity to outliers and multicollinearity issues. It can handle high latitude, discrete or continuous data without data normalization and is easy to implement and parallel. However, several limitations exist, e.g., it is difficult to make predictions beyond the range of training data, and there will be an over-fitting issue on some regression problems with high noise. The training efficiency will reduce with increasing memory occupation when the number of decision trees is large (Geurts et al., 2006).

Compared with traditional tree-based models (e.g., random forest), ERT has a stronger randomness which randomly selects a feature subset at each node split and randomly obtains the optimal branch attribute and threshold. This helps to create more independent decision trees, further reducing model variance and improving training accuracy (Geurts et al., 2006). The STET model has been successfully applied in estimating high-quality surface $O_3$ in our previous study (Wei et al., 2022a). It is thus extended here to regress the nonlinear conversion relationships between ground-based measurements and the main predictors and auxiliary factors for other species of gaseous pollutants. For surface $NO_2$, the STET model was applied to the main variables of the satellite tropospheric $NO_2$ column, modelled surface $NO_2$ mass, and $NO_x$ emissions, together with ancillary variables of the previously mentioned meteorological, surface, and population variables (Equation 1). For surface $SO_2$ (Equation 2) and CO (Equation 3), modelled surface $SO_2$ and CO concentrations and $SO_2$ and CO emissions were used as main predictors along with the same auxiliary variables as $NO_2$ to construct the STET models separately:"

Line 132-137: You have to first describe the product you are creating. Then you can discuss the data quality. A simple reference to a previous paper is not enough.

Response: We have moved this part of the discussion on data quality after the product description in the revision, as you suggested.

Line 140: You first have to show your product, on a map. Have you captured the geographical spread properly? then you should show seasonal maps. have you captured the seasonality properly? then you should discuss other patterns, for e.g. the COVID lockdown effect. On maps.

Response: We admit that the logic of the article is somewhat confusing. We have adjusted the order by first showing and discussing the mapping results in the revision, according to your suggestions.

The scatter plots of Figure 1 cannot be the first result you present of your work. In your title you claim "seamless mapping". This should be they first thing you demonstrate.
Response: We revised this section by first demonstrating the "seamless mapping" of our product, then discussing the validation results, as you suggested.

Lines 143-144: What is this dataset? is it open source? where is it being used?
Response: ChinaHighAirPollutants (CHAP, available at https://weijing-rs.github.io/product.html) is the name of a series of public high-resolution, high-quality data sets of a variety of ground-level air pollutants for China developed by our team, including the three gaseous pollutants generated in this study. We have clarified this in the revision.

Lines 146-148: The methodology was poorly described, hence it is impossible for the reader to understand and assess the validity of these numbers.
Response: We have re-described the methodology in more detail according to your suggestions (see the response to your previous comment).

Line 155: Are these spatial correlations? temporal correlations? after deseasonalisation? before? There are too many issues left un-described for the reader to assess your writings.
Response: These are spatial correlations between our original predictions and ground measurements of three gaseous pollutants using the out-of-station cross-validation method. We have clarified this in the revision.

Line 161: Table 1 in not discussed anywhere in the text.
Response: It is now Table S1 in the revision.

Lines 164-165: You have to show the geographical spread of your stations on a map.
Response: We have added a figure (Figure 1) showing the geographical locations of ground-based monitoring stations in the revision.

Also, you have to show how you separated the stations for the training and the validation parts.
Response: The out-of-station ten-cross validation approach randomly divides the ground monitoring stations into ten subsets, of which data samples from nine subsets are used for model training, and the remaining subset for model validation. It runs 10 times, in turn, to ensure that data from all monitors are tested. This procedure generates independent training samples and test samples made in different locations, used to indicate the spatial prediction ability of the model in areas where ground-based measurements are unavailable. We have clarified this in the revised Section 3.3.1.

Lines 165-167: Who performs this QC? how is this QC performed? a proper reference should

be described here.

Response: Here, the QA/QC was performed in 2018 by the China National Environmental Monitoring Centre (CNEMC) with updated technical specifications for the operation and quality control of ambient air quality continuous automated monitoring of gaseous pollutants, including the improvement of sampling flow calibration of monitoring instruments, flow calibration of dynamic calibrators, and revision of precision/accuracy review and data validity judgment (HJ 818-2018) (MEE, 2018b). We have clarified this in the revision (see revised Section 3.3.1).

Lines 173-177: So, for all 3 species, 80-83% of the stations shown an CVR2 > 0.6. Hence, either all three capture or all three do not capture the daily variability. RMSE absolute values give no information to the reader, if the typical levels for surface pollutants is not known. Furthermore, these levels vary greatly depending on the location of the station. This part is also poorly written.

Response: We agree. We have replaced RMSE absolute values with normalized RMSE (NRMSE) values to better describe the estimated uncertainties for gaseous pollutants in the revision. Also, we have rephrased this part as follows:

"In general, our model works well at the site scale, with average CV-$R^2$ values of 0.77, 0.72, and 0.72, and NRMSE values of 0.25, 0.43, and 0.26 for surface $NO_2$, $SO_2$, and CO, respectively. In addition, approximately 93%, 80%, and 84% of the stations had at least moderate agreements (CV-$R^2$ > 0.6) between our estimates and ground measurements. Except for some scattered sites, the estimation uncertainties were generally less than 0.3, 0.5, and 0.3 in more than 80%, 77%, and 76% of the stations for the above three gaseous pollutant species, respectively."

Lines 182-183: All monitoring stations or just the validation stations? this is not explained properly.

Response: Here refers to the validation stations, now explained in the revision.

Lines 188-190: Figure 3 does not demonstrate what you claim here.

Response: We have deleted this sentence from the revision.

Lines 199-201: How does Figure 4 compare to the input datasets? if you plot the CAMS reanalysis set, and their difference, by how much do they differ? do they auto-correlate? This is a very important point that needs addressing.

Response: We have compared our results with the input CAMS reanalysis dataset by plotting the spatial distributions and calculating their correlations and differences against surface observations in the revision:

"In addition, reanalysis data do not simulate surface masses of gaseous pollutants well, underestimating them compared to our results and ground-based observations in China (Figure S2). This is especially so for $SO_2$, where high-pollution hot spots are easily misidentified. Validation illustrates that our regressed results for surface $NO_2$, $SO_2$, and CO

agree better with ground measurements than modelled results (slopes are close to 1, and correlations > 0.93), 1.9–6.4 times stronger in slope, 1.3–3.5 times higher in correlation, but 5.9–7.7 times smaller in differences (Figure S3). This shows that our model can take advantage of big data to significantly correct and reconstruct gaseous simulation results via data mining using machine learning."

Line 203: Average over all of China? average during a year?
Response: It is averaged over all of China during the period 2013–2020. We have clarified this in the revision.

Line 204: Daily data satellite observations?
Response: Corrected.

Line 220: How does meteorology affect SO2 and CO? Furthermore, NO2 is affected by the sunlight and the photochemical reactions taking place, this is not a meteorological condition.
Response: The lowest levels of gaseous pollutants in summer are due to favorable meteorological conditions, e.g., abundant precipitation and high air humidity conducive to flushing and scavenging different air pollutants (Yoo et al., 2014). We agree with you that $NO_2$ is also affected by strong sunlight and high temperature that accelerate the photochemical reactions of $NO_2$ loss (Shah et al., 2020). We have clarified this and cited the references in the revision.

Section 3.2.2: Many studies have been performed over China for the COVID lockdown effects on air quality. How does your work add to the knowledge already in existence?
Response: We agree with you that there are many studies focused on the effects of the COVID-19 epidemic (WHO, 2020) on air quality. Most of them were done using ground-based observations (Huang et al., 2020; T. Su et al., 2020), tropospheric gas columns (Field et al., 2021; Levelt et al., 2022), or retrieved surface masses (Ling and Li, 2021; Cooper et al., 2022). The resulting conclusions could be affected by insufficient spatial representation due to the uneven distribution of ground monitors or a large number of missing values in space due to the influence of clouds. The unique advantage of our seamless day-to-day gaseous pollutant dataset can make up for these shortcomings, allowing us to more accurately and quantitatively assess the changes in gaseous pollutants during the epidemic. In addition, most previous studies have focused mainly on changes during the lockdown, with little attention paid to the recovery. Our study quantified this. We found that surface CO was the first to return to its historical level within the fifth week after the Lunar New Year in 2020, about twice faster as surface $NO_2$ and $SO_2$ levels. This is attributed to more home cooking and enhanced atmospheric oxidation. We have clarified this in the revised Section 3.2.1.

Lines 254-255: Since your dataset is a seamless 10x10km map of all three pollutants, the availability of ground based monitoring stations should not affect you. This is the main point of your work. Hence, I fail to see why you limit yourself to the locations where ground-based monitoring exists.
Response: We agree with you. We have removed such descriptions and redrew the figure by

including western China. We discussed the temporal variations of gaseous pollutants for the whole of China in the revision.

Line 258: The 2020 levels were affected by COVID.
Response: We agree with you and have clarified this in the revision.

Line 264: For which species? is 1.1microgram a significant amount?
Response: We have rephrased the sentence as "Most of China showed significant decreasing trends, with average annual rates of 0.23 µg/m$^3$, 2.01 µg/m$^3$, and 0.05 mg/m$^3$ for surface NO$_2$, SO$_2$, and CO, respectively" in the revision. A decile rate of 1.1 µg/m$^3$ is a significant amount because the average surface NO$_2$, SO$_2$, and CO levels in China were only $20.3 \pm 4.7$ µg/m$^3$, $16.2 \pm 7.7$ µg/m$^3$, and $0.86 \pm 0.22$ mg/m$^3$ during the period 2013–2020.

Lines 269-270: What is the reason for this?
Response: Increasing trends of surface NO$_2$ were found in Ningxia and Shanxi Provinces in central China due to increased traffic emissions and new coal-burning power plants in underdeveloped areas without strict regulations on NOx emissions (van der A et al., 2017; Maji and Sarkar, 2020; Li et al., 2022a). We have clarified this in the revision.

Line 299: "Level 2 limitation" What does this mean?
Response: Level 2 limitation refers to the secondary concentration limit of ambient air quality standards formulated by China (GB3095-2012). In the revised version, we have replaced it with the newly defined air quality guidelines (AQG) formulated by WHO in 2021, according to the suggestion of Reviewer 1.

Line 300: Exposure to what?
Response: Exposure to three gaseous pollutants, clarified in the revision.

Line 303: This phrase contradicts what you stated before and Figure 9.
Response: We have deleted this phrase from the revision.

Line 313: "Regionally" For which regions?
Response: Here we mean the regional scale.

Lines 314-316: Since Figure 10 refers to three regions, why is only BTH discussed here?
Response: The BTH region was the only region with high NO$_2$, SO$_2$, and CO exposure risks. The other two regions had no such risks. We have added "YRD and PRD had no high risks of exposure to the three gaseous pollutants …" in the revision.

Section 3.3: This section is overall poor. There is no mention as to how the comparison was made.
Response: Here, only those studies applying the same out-of-sample cross-validation approach against ground-based measurements collected from the same CNEMC network were selected. This makes our comparison fair. We have clarified this in the revision.

0.125x0.125 in degrees is roughly 12.5x12.5 km which is very close to your 10x10km. Hence, you cannot claim that the other datasets have "low spatial resolutions".

Response: We have rephrased the sentence as "Most generated surface $NO_2$ datasets had numerous missing values in space limited by direct OMI satellite observations at spatial resolutions from $0.125° \times 0.125°$ to $0.25° \times 0.25°$" in the revision.

Line 336: By how much? compared to your quality?

Response: Surface $SO_2$ estimated from an $SO_2$ emission inventory and surface CO from Measurement of Pollution in the Troposphere (MOPITT) and TROPOMI retrievals have a lower data quality, with smaller $R^2$ values by 12–57% and larger RMSE values by 41–47% against ground measurements compared to ours (D. Liu et al., 2019; R. Li et al., 2020; Y. Wang et al., 2021). We have added this in the revision.

Line 344: Where? add link.

Response: We have added the links for the datasets of the three surface gaseous pollutants.

Lines 348-354: Have all these studies based their results directly on your database? is this implied here?

Response: Yes, the results of all the studies listed in this section were obtained using the three gaseous pollutant datasets (i.e., $NO_2$, $SO_2$, and CO) generated in this study. We have clarified this in the revision.

---

## Author Response (AR2)

This study constructed 10 km daily surface concentrations of three major ambient pollutant gases, (NO2, SO2, and CO) across China from 2013 to 2020 based on the machine learning method. They also examined the variations in the pollutants in recent years. Other reviews have raised many good comments and suggestions. I still have some comments that can be addressed before publication.

My concern is the predictors that were used to train the model and predict pollutants' concentrations. The surface measurements from MEE are mainly over eastern China. Only NO2 from satellite products were used to supplement the surface data, and it also has large uncertainties for deriving surface information. For SO2 and CO, they applied model simulations to solve the lack of data in China. However, the model data also have large biases over the regions without observations. I understand the biases due to the lack of real-time data can not be easily solved. Some studies also used model results to train the machine model (e.g., Li et al., 2022). The authors should discuss the limitation in detail and provide the caveat of the results.

Response: Thanks for your suggestion and we have discussed the limitations, pointed the caveat of the results, and cited the reference in the revised conclusion as below:

"our estimated surface $SO_2$ and CO concentrations should have larger uncertainties than those of $NO_2$ since model simulations stead of satellite retrievals are supplemented during modelling to compensate for the lack of data in China. However, these data often have large biases in the remote regions with few observations in western China (Li et al., 2022), as the surface measurements from MEE are mainly over eastern China."

Specific comments:
For the meteorological data, winds at 850 hPa are commonly used to represent the transport of air pollutants and the pressure system in the mid-troposphere is also important for the pollution accumulation. The prediction could be more accurate if more meteorological parameters are included or at least discuss the potential bias without considering these factors.

Response: The meteorological system is complex, which can pose varying impacts on air pollutants, and we agree with you that considering more meteorological factors (e.g., winds at 850 hPa and pressure system in the mid-troposphere) may obtain more accurate estimates. Considering that we have included the variables related to winds and pressure near the ground, e.g., 10-m winds and surface pressure (1000 hPa), also the vertical distributions and variations of air pollutants in the boundary layer (i.e., boundary layer height), making such subtle adjustments will not greatly change the accuracy. Nevertheless, we have discussed this limitation and potential bias in the revised Conclusion according to your suggestion as follows:

"more parameters describing the complex meteorological system (e.g., winds at 850 hPa and the pressure system in the mid-troposphere) need to be considered in developing more powerful artificial intelligence models, which could be helpful in improving the accuracy of air pollutant retrievals."

Line 55: Many latest studies also modeled recent variation of air pollutants in China (e.g., Gao et al., 2022; Yang et al., 2022).
Response: We have summarized and cited these related studies in the revised Introduction.

References:
Li, H., Yang, Y., Wang, H., Wang, P., Yue, X., and Liao, H.: Projected Aerosol Changes Driven by Emissions and Climate Change Using a Machine Learning Method, Environ. Sci. Technol., 56, 7, 3884–3893, https://doi.org/10.1021/acs.est.1c04380, 2022.

Gao, J., Yang, Y., Wang, H., Wang, P., Li, H., Li, M., Ren, L., Yue, X., and Liao, H.: Fast climate responses to emission reductions in aerosol and ozone precursors in China during 2013–2017, Atmos. Chem. Phys., 22, 7131–7142, https://doi.org/10.5194/acp-22-7131-2022, 2022.

Yang, Y., Ren, L., Wu, M., Wang, H., Song, F., Leung, L. R., Hao, X., Li, J., Chen, L., Li, H., Zeng, L., Zhou, Y., Wang, P., Liao, H., Wang, J., and Zhou, Z.-Q.: Abrupt emissions reductions during COVID-19 contributed to record summer rainfall in China, Nat. Commun., 13, 959, https://doi.org/10.1038/s41467-022-28537-9, 2022.